# A transcriptional network governing ceramide homeostasis establishes a cytokine-dependent developmental process

Ruiqi Liao [1], Abiola Babatunde [1], Stephanie Qiu[1], Hamsini Harikumar[1], Joshua J. Coon [2,3,4], Katherine A. Overmyer [2,3], Yusuf A. Hannun[5,6], Chiara Luberto[7] & Emery H. Bresnick [1] ✉

Transcriptional mechanisms controlling developmental processes establish and maintain proteomic networks, which can govern the levels of intracellular small molecules. Although dynamic changes in bioactive small molecules can link transcription factor and genome activity with cell state transitions, many mechanistic questions are unresolved. Using quantitative lipidomics and multiomics, we discover that the hematopoietic transcription factor GATA1 establishes ceramide homeostasis during erythroid differentiation by regulating genes encoding sphingolipid metabolic enzymes. Inhibiting a GATA1-induced sphingolipid biosynthetic enzyme, delta(4)-desaturase, or disrupting ceramide homeostasis with cell-permeable dihydroceramide or ceramide is detrimental to erythroid, but not myeloid, progenitor activity. Coupled with genetic editing-based rewiring of the regulatory circuitry, we demonstrate that ceramide homeostasis commissions vital stem cell factor and erythropoietin signaling by opposing an inhibitory protein phosphatase 2A-dependent, dual-component mechanism. Integrating bioactive lipids as essential components of GATA factor mechanisms to control cell state transitions has implications for diverse cell and tissue types.

With the emergence of advanced omic technologies, amalgamating datasets (multiomics) constitutes a powerful approach to dissecting biological problems, including how transcriptional mechanisms establish and maintain complex genetic networks to control cell state transitions. While transcriptomic data often provide a wealth of clues, transcriptomic and proteomic data can be discordant, highlighting the need to extend beyond RNA-centric inferences to accrue more decisive evidence. This is especially applicable when RNA or protein expression data suggest the involvement of small molecules, *e.g.*, metabolites, metals, and lipids, in transcription factor-instigated regulatory networks. The contributions of endogenous small molecules are often not considered while deducing biological and pathological mechanisms from RNA- and protein-based omic data. This issue is particularly critical when small molecules possess bioactivities that influence cell function.

Our multiomic studies to elucidate hematopoietic GATA transcription factor mechanisms discovered target genes encoding metabolic enzymes and membrane transporters that regulate intracellular small molecule levels[1–4]. In erythroblasts that differentiate into erythrocytes, GATA1 induces heme synthesis by activating enhancers in

[1]Wisconsin Blood Cancer Research Institute, Department of Cell and Regenerative Biology, Carbone Cancer Center, University of Wisconsin School of Medicine and Public Health, Madison, WI, USA. [2]Department of Biomolecular Chemistry, National Center for Quantitative Biology of Complex Systems, University of Wisconsin School of Medicine and Public Health, Madison, WI, USA. [3]Morgridge Institute for Research, Madison, WI, USA. [4]Department of Chemistry, University of Wisconsin-Madison, Madison, WI, USA. [5]Department of Medicine, Stony Book University, Stony Brook, NY, USA. [6]Northport Veterans Affairs Medical Center, Northport, NY, USA. [7]Department of Physiology and Biophysics, Stony Brook University, Stony Brook, NY, USA. ✉e-mail: ehbresni@wisc.edu

*Alas2*, encoding the rate-limiting heme biosynthetic enzyme[1]. Beyond its roles as an enzyme cofactor and hemoglobin component, heme facilitates or antagonizes GATA1, in part by binding and instigating the degradation of the transcriptional repressor BACH1[1,3]. The GATA1/heme-activated gene *Slc30a1* encodes the major zinc exporter in erythroid cells. GATA1 induces expression of the zinc importer *Slc39a8* and exporter *Slc30a1*, elevating intracellular zinc to confer erythroblast survival. Upon differentiation, *Slc39a8* expression declines, and *Slc30a1* expression is sustained, reducing intracellular zinc. Lowering Slc30a1 increases intracellular zinc and promotes differentiation[2].

Our analysis of genes encoding >450 solute carrier (SLC) transporters in GATA1 genetic complementation and GATA2 enhancer-mutant systems revealed GATA factor regulation of >50 SLC genes, including *Slc29a1* encoding an equilibrative nucleoside transporter that utilizes adenosine as a substrate[4]. Depleting Slc29a1 impaired murine erythroblast survival and differentiation ex vivo, while targeted ablation of *Slc29a1* in mouse erythroblasts attenuated erythropoiesis and erythrocyte regeneration in vivo[4]. Human SLC29A1 deficiency is associated with erythroid differentiation and erythrocyte defects[5]. Thus, GATA factor-regulated genes can encode proteins that process and utilize small molecules, and small molecules exert vital functions in GATA factor networks. Given the vast small molecule ensembles in cells, many small molecule-dependent mechanisms governing genome function and cell state transitions remain undefined.

Utilizing quantitative lipidomic and multiomic approaches with a GATA1 genetic complementation system (G1E-ER-GATA1), we discovered that GATA1 regulates genes and proteins mediating sphingolipid metabolism. Bioactive sphingolipids include ceramides, sphingosine, and sphingosine-1-phosphate that control processes including proliferation, survival, differentiation, inflammation, signaling, and metabolism[6-8]. Pharmacological inhibition of DEGS1, which mediates de novo sphingolipid synthesis, promotes hematopoietic stem cell (CD34$^+$) self-renewal[9]. As a component of its signaling mechanism, TNFα promotes sphingomyelin hydrolysis to yield ceramides (Cer) and inhibits erythroid differentiation from human CD34$^+$ cells[10]. How sphingolipid biosynthesis is controlled during hematopoiesis and how signaling lipids operate within hematopoietic-regulatory networks are not established.

Herein, we conducted mechanistic analyses in the G1E-ER-GATA1 system, as well as primary murine and human erythroid cells, to test whether bioactive sphingolipids function in GATA1 networks. Disrupting ceramide homeostasis by inhibiting a GATA1-induced enzyme or with cell-permeable ceramides strongly impaired erythroid, but not myeloid, progenitor function. Analyses using pharmacological inhibitors and activators, and genetic editing-based rewiring of the regulatory circuitry, revealed that disrupted ceramide homeostasis opposes cytokine signaling pathways in erythroid cells via an acute mechanism targeting AKT and ERK and a delayed mechanism targeting the JAK/STAT pathway, both involving protein phosphatase 2 A (PP2A) activation. These findings illustrate how a transcriptional mechanism establishes ceramide homeostasis to commission cytokine signaling that controls cellular proliferation, differentiation, and erythrocyte development.

## Results

### GATA1 establishes ceramide homeostasis during erythroid differentiation

GATA1 regulates genes encoding small molecule metabolic enzymes and transporters[1,2,4,11]. Mining our existing multiomic datasets revealed that multiple genes and proteins mediating sphingolipid metabolism are GATA1-regulated. However, mechanistic and biological links between GATA factor and sphingolipid mechanisms have not been described. Sphingolipid metabolism in mammalian cells is mediated by enzymatic reactions occurring in the endoplasmic reticulum (ER) and Golgi membranes[8,12] (Fig. 1a). To investigate whether sphingolipid-derived signaling lipids are essential components of GATA1-instigated regulatory networks, we assessed whether GATA1 controls genes encoding sphingolipid metabolic enzymes using a GATA1 genetic complementation system. G1E-ER-GATA1 cells are GATA1-null proerythroblast-like cells stably expressing a β-estradiol-induced allele encoding GATA1 fused to the estrogen receptor hormone-binding domain[13,14]. Comparing the G1E-ER-GATA1 cell transcriptome[1] and proteome[2], with or without active ER-GATA1, revealed differentially expressed components mediating sphingolipid biosynthesis[1,2] (Fig. 1b and c). *Degs1* encodes dihydroceramide desaturase (DES) that converts dihydroceramides to ceramides (Fig. 1a). GATA1 increased *Degs1* mRNA and protein expression 12- ($p = 0$) and 4.6-fold ($p = 3.4e-4$), respectively (Fig. 1b and c). RT-qPCR confirmed that GATA1 increased *Degs1* RNA 2.7- ($p = 0.022$) and 6.7-fold ($p < 0.0001$) at 24 and 48 h post-β-estradiol treatment, respectively (Fig. 1d). As a control for GATA1-mediated erythroid differentiation, we analyzed *Gata2* expression, which rapidly declined; expression of the constitutively expressed *Eif3k* gene was constant. Semi-quantitative Western blotting to detect DEGS1 yielded similar results (Fig. 1e and f). ATAC-seq demonstrated that GATA1 induced chromatin accessibility at the murine *Degs1* promoter[3], and GATA1 occupied intronic sites in G1E-ER-GATA1 cells and primary murine erythroblasts. In human peripheral blood-derived erythroblasts (PBDE), GATA1 occupied the promoter and sequences flanking *DEGS1*[15]. As the GATA1-occupied sequences contained one or more motifs mediating GATA1 binding (WGATAR)[11,16,17], *Degs1* is a GATA1 target gene (Supplementary Fig. 1a).

To determine if GATA1-mediated regulation of sphingolipid metabolic enzyme expression alters sphingolipid levels during erythroid maturation, quantitative lipidomics was conducted in G1E-ER-GATA1 cells cultured with or without β-estradiol for 24 and 48 h. β-estradiol increased total ceramide (Cer) level 33% ($p = 0.036$) and 32% ($p = 0.038$) at 24 and 48 h, respectively (Fig. 1g). Total dihydroceramide (dhCer) level was unaffected. Increased Cer without reduced dhCer is consistent with increased conversion of dhCer to Cer resulting from GATA1-induced *Degs1* expression coupled with increased flux in de novo sphingolipid synthesis to sustain dhCer levels. Sphingosine (Sph) and sphingosine-1-phosphate (Sph-1P) increased 3.9- ($p < 0.0001$) and 2.9-fold ($p < 0.0001$), respectively, consistent with GATA1-induced *Sphk1* and *Sphk2*, which encode sphingosine kinases. GATA1-mediated induction of Sph-1P provides a mechanistic explanation for how erythrocytes generate and elaborate Sph-1P to maintain vascular homeostasis[18-23].

Analysis of Cer and dhCer species revealed that GATA1 elevated multiple Cer species, including the most abundant species C16-Cer. GATA1 elevated specific dhCer species, including C16-dhCer, while reducing others e.g., C24-dhCer (Fig. 1h, Supplementary Data 1). An alternative strategy utilizing discovery relative-quantitation lipidomics revealed multiple GATA1-induced Cer species (Supplementary Fig. 1b, Supplementary Data 2). The differential regulation of dhCer with variable acyl chain lengths is commensurate with the differential regulation of ceramide synthase genes. GATA1 induces CERS6, which prefers C14 and C16 fatty acid chains, whereas GATA1 represses CERS2, which prefers C20-C26 chains[24,25]. These studies demonstrated that GATA1 regulation of sphingolipid metabolic genes establishes and maintains Cer homeostasis.

### Ceramide homeostasis requirement for erythroblast function and erythrocyte development

To assess the significance of GATA1-induced *Degs1* expression, we pharmacologically inhibited DES with 4-hydroxyphenyl retinamide (4HPR)[26] in G1E-ER-GATA1 cells and quantified sphingolipids with quantitative lipidomics. 4HPR increased dhCer 7.8- ($p < 0.0001$) and 14-fold ($p = 0.0016$) at 24 and 48 h, respectively, while Cer decreased 1.7 ($p = 0.0003$) and 1.9-fold ($p = 0.0006$), respectively (Fig. 2a). Analysis of dhCer species revealed that 4HPR elevated most species, with

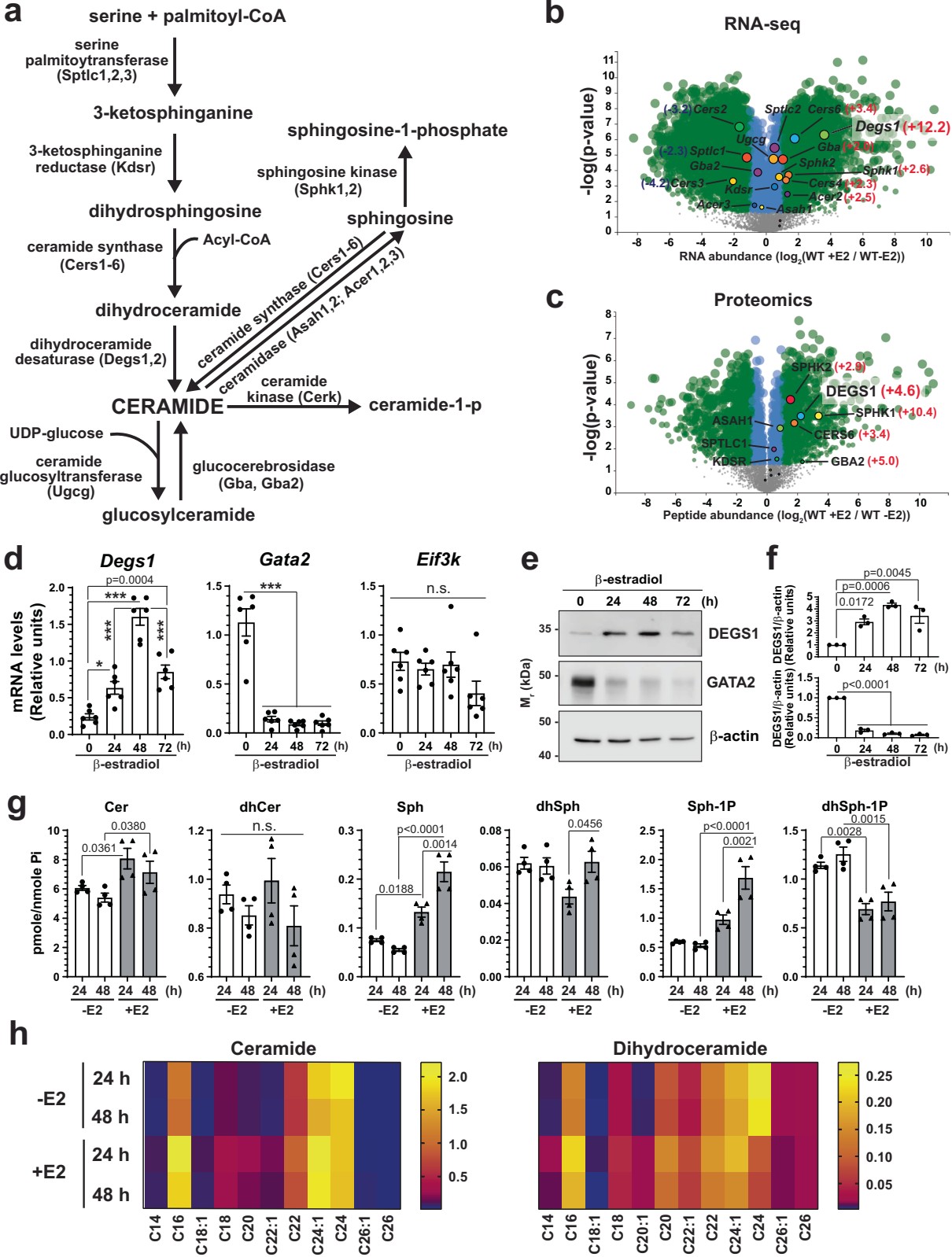

C24-dhCer increasing to the greatest extent. 4HPR decreased most ceramide species (Fig. 2b, Supplementary Data 1). To determine if 4HPR-elicited changes in dhCer and Cer species result from differential regulation of ceramide synthases, we quantified RNA expression of *Cers* genes using RT-qPCR. In G1E-ER-GATA1 cells, only *Cers2* and *Cers6* are abundantly expressed (Supplementary Fig. 2a). 24 h 4HPR treatment did not alter *Cers2* and *Cers6* expression (Supplementary Fig. 2b),

suggesting that 4HPR disrupts ceramide homeostasis without altering *Cers* gene expression.

We assessed whether Cer homeostasis is a determinant of erythroblast function and/or differentiation. 4HPR strongly reduced live cell numbers and viability of undifferentiated (-β-estradiol) or differentiated (+β-estradiol) G1E-ER-GATA1 cells 24 h post-treatment (Fig. 2c and d), suggesting that DES mediates proliferation and/or survival.

**Fig. 1 | GATA1 establishes ceramide homeostasis by regulating sphingolipid metabolic enzyme genes. a** Diagram illustrating sphingolipid metabolic pathways and the respective enzymes. **b** Volcano plot depicting GATA1-regulated RNAs involved in sphingolipid biosynthesis. RNA-seq data from 3 biologically independent samples were log$_2$ transformed and $p$ values were calculated by two-tailed unpaired Student's test. **c** Volcano plot depicting GATA1-regulated proteins involved in sphingolipid biosynthesis. Proteomic data from 3 biologically independent samples were log$_2$ transformed and $p$ values were calculated by two-tailed unpaired Student's test. **d** RT-qPCR validation of GATA1-dependent regulation of *Degs1* mRNA during β-estradiol-induced differentiation of G1E-ER-GATA1 cells ($n = 6$ biologically independent samples, mean ± SEM). $p$ values were calculated using one-way ANOVA followed by Tukey's multiple comparisons test. *$p = 0.0219$,

***$p < 0.0001$, n.s., not significant. **e** Representative Western blots showing GATA1-dependent regulation of DEGS1 protein during β-estradiol-induced differentiation of G1E-ER-GATA1 cells. **f** Quantitation of Western blots in (**e**). Band intensities of DEGS1 and GATA2 were normalized to the corresponding β-actin band ($n = 3$ biologically independent samples, mean ± SEM). $p$ values were calculated using one-way ANOVA followed by Tukey's multiple comparisons test. **g** Lipidomic analysis to quantify sphingolipid levels during β-estradiol-induced differentiation of G1E-ER-GATA1 cells ($n = 4$ biologically independent samples, mean ± SEM). The levels of lipids were normalized to the level of inorganic phosphate (Pi). $p$ values were calculated by two-tailed paired Student's test. E2, β-estradiol. **h** Heatmaps depicting levels of individual species of Cer and dhCer from (**g**). Source data are provided as a Source Data file.

Flow cytometry-based Annexin V staining analysis revealed that 4HPR increased the percentage of early apoptotic cells by 5.8-fold ($p < 0.0001$) and late apoptotic cells by 6.0-fold ($p < 0.0001$) (Supplementary Fig. 2c and d). Similarly, 4HPR strongly decreased live cell numbers and viability of human primary erythroblasts 24 h post-treatment (Fig. 2e) and increased the percentage of early and late apoptotic cells by 6.0-fold ($p < 0.0001$) and 18-fold ($p < 0.0001$), respectively (Supplementary Fig. 2e and f), suggesting that this mechanism is conserved between mouse and human. ER-GATA1 activation induces synchronous erythroid maturation within 48 h, which can be monitored by the emergence of pink/red color, indicating hemoglobinization. 4HPR-treated cells did not accumulate hemoglobin after 48 h of β-estradiol treatment (Fig. 2f), suggesting that 4HPR impairs maturation. Flow cytometric analyses revealed that 4HPR reduced the percentage of CD71$^+$Ter119$^-$ and increased the percentage of CD71$^-$Ter119$^-$ cells 48 h post-treatment in G1E-ER-GATA1 and primary murine fetal liver cells (Supplementary Fig. 2g–j), indicating that 4HPR inhibits maturation. 4HPR reduced GATA1-mediated induction of the erythroid-specific genes *Hba-a1*, *Hbb-b1*, *Alas2*, and *Slc4a1*[11] by 50–70% without affecting GATA1-induced repression of genes (*Gata2* and *Kit*)[14,27] that occurs early in erythroid differentiation (Fig. 2g).

To determine if Cer homeostasis is a determinant of primary erythroid progenitor function, we isolated lineage-depleted (Lin$^-$) E14.5 murine fetal liver cells[28] and performed colony forming unit (CFU) assay with or without DES inhibition. 4HPR nearly eliminated erythroid colonies (CFU-E declined 4.8-fold, $p < 0.0001$, and BFU-E declined 4.8-fold, $p = 0.0008$), without affecting myeloid colony (CFU-GM) numbers (Fig. 3a). Wright-Giemsa staining of cells isolated from colonies revealed that 4HPR depleted immature erythroblasts (Fig. 3b). Thus, myeloid, but not erythroid, progenitors can tolerate DES inhibition.

As an alternative strategy to assess the relationship between Cer homeostasis and progenitor activity, we treated Lin$^-$ E14.5 fetal liver cells with C6-dihydroceramide (C6-dhCer) or C6-ceramide (C6-Cer) to analyze the direct actions of ceramides and conducted CFU assays. C6-dhCer and C6-Cer reduced the number of CFU-E 43 and 96-fold ($p < 0.0001$) and BFU-E 9.5 and 30-fold ($p < 0.0001$), respectively. Both C6-dhCer and C6-Cer elevated CFU-GM (1.9-fold, $p = 0.0011$) (Fig. 3c). Wright-Giemsa staining of cells isolated from colonies revealed that C6-dhCer- and C6-Cer-treated samples generated predominantly neutrophils. Erythroblasts were abundant in control plates (Fig. 3d). Treating G1E-ER-GATA1 or hi-WT cells (HOXB8-immortalized wild type, fetal liver-derived myeloid progenitors)[29,30] with increasing amounts of C6-dhCer and C6-Cer revealed a greater sensitivity of erythroid vs. myeloid cells to dhCer and Cer (Supplementary Fig. 3). Thus, disrupting Cer homeostasis to yield excessive dhCer or Cer is disproportionately deleterious to erythroid progenitors. As 4HPR-mediated DES inhibition strongly increased (7.8–14 fold) dhCer, while decreasing Cer by 1.7-1.9 fold, the adverse impact of 4HPR on erythroid progenitor function may reflect the dhCer accumulation or more complex alterations in the dynamic regulation of both species.

## Ceramide homeostasis dictates cytokine signaling efficacy

To elucidate mechanisms underlying the erythroid cell hypersensitivity to (dh)Cer, we asked if disrupting Cer homeostasis alters the integrity of cytokine signaling systems that mediate erythroid progenitor survival, proliferation, and differentiation. Stem cell factor (SCF) and erythropoietin (Epo) signaling control erythropoiesis through independent and intertwined actions[31–33]. 4HPR-mediated DES inhibition strongly reduced SCF-induced p-AKT and p-ERK1/2 and Epo-induced p-AKT, p-ERK1/2, and p-STAT5 (Fig. 4a–c). C6-dhCer or C6-Cer reduced SCF-induced p-AKT and p-ERK1/2 and Epo-induced p-AKT, p-ERK1/2, and p-STAT5 (Fig. 4d–f), indicating that SCF and Epo signaling are sensitive to the dhCer and Cer status of cells. Since DES inhibition by 4HPR caused accumulation of dhCer at the expense of Cer, we asked if supplementing the cells with Cer rescues 4HPR-dependent inhibition of cytokine signaling. C6-Cer treatment further reduced Epo-induced p-AKT, p-ERK, and p-STAT5 in 4HPR-treated cells (Supplementary Fig. 4a and b), indicating that both dhCer and Cer are deleterious to cytokine signaling. To determine if Cer homeostasis impacts human cytokine signaling, we treated primary human erythroblasts with 4HPR, C6-dhCer, or C6-Cer and quantified SCF and Epo signaling. All treatments strongly reduced SCF and Epo-induced p-ERK1/2 and p-STAT5 (Fig. 4g–i). Thus, the Cer homeostasis requirement for cytokine signaling systems constitutes a conserved mechanism. As ceramide homeostasis promoted or was essential for cytokine signaling, we refer to this mechanism as ceramide homeostasis-dependent commissioning of signaling.

To dissect the mechanism, we asked if ceramide homeostasis impacts cytokine receptor expression or activation. 4HPR treatment (4 h) did not alter RNA expression of *Kit* and *Epor*, encoding receptors for SCF and Epo, respectively (Supplementary Fig. 4c). By contrast, cytokine-dependent phosphorylation of KIT and EPOR was significantly reduced (Fig. 4j and k). C6-dhCer and C6-Cer also decreased cytokine-induced p-KIT and p-EPOR, with EPOR phosphorylation reduced to a greater extent than KIT by C6-Cer (Fig. 4l and m). These results indicate that ceramide homeostasis confers cytokine-dependent receptor activation. We asked if disrupting Cer homeostasis and the resulting inhibition of cytokine signaling impacts nuclear processes. Epo-activated p-AKT induces GATA1 phosphorylation at S310[34–36]. Epo treatment of Epo-starved G1E-ER-GATA1 cells induced pS310-GATA1 in a dose-dependent manner (Supplementary Fig. 4d and e). C6-Cer treatment before Epo attenuated Epo-induced GATA1 phosphorylation, shifting the dose-response curve to the right. By reducing Epo potency, C6-Cer restricts signal-dependent transcription factor targeting (Supplementary Fig. 4d and e).

## Genetically rewiring ceramide-cytokine receptor circuitry

Since ceramides directly bind and activate the protein phosphatase PP2A[37], and the activation is stereospecific and regulated by phosphatidic acid, we asked if PP2A activation mediates Cer-dependent repression of cytokine signaling. 4HPR, C6-dhCer, and C6-Cer suppressed SCF- and Epo-induced p-AKT, p-ERK1/2, and p-STAT5 (Fig. 4)

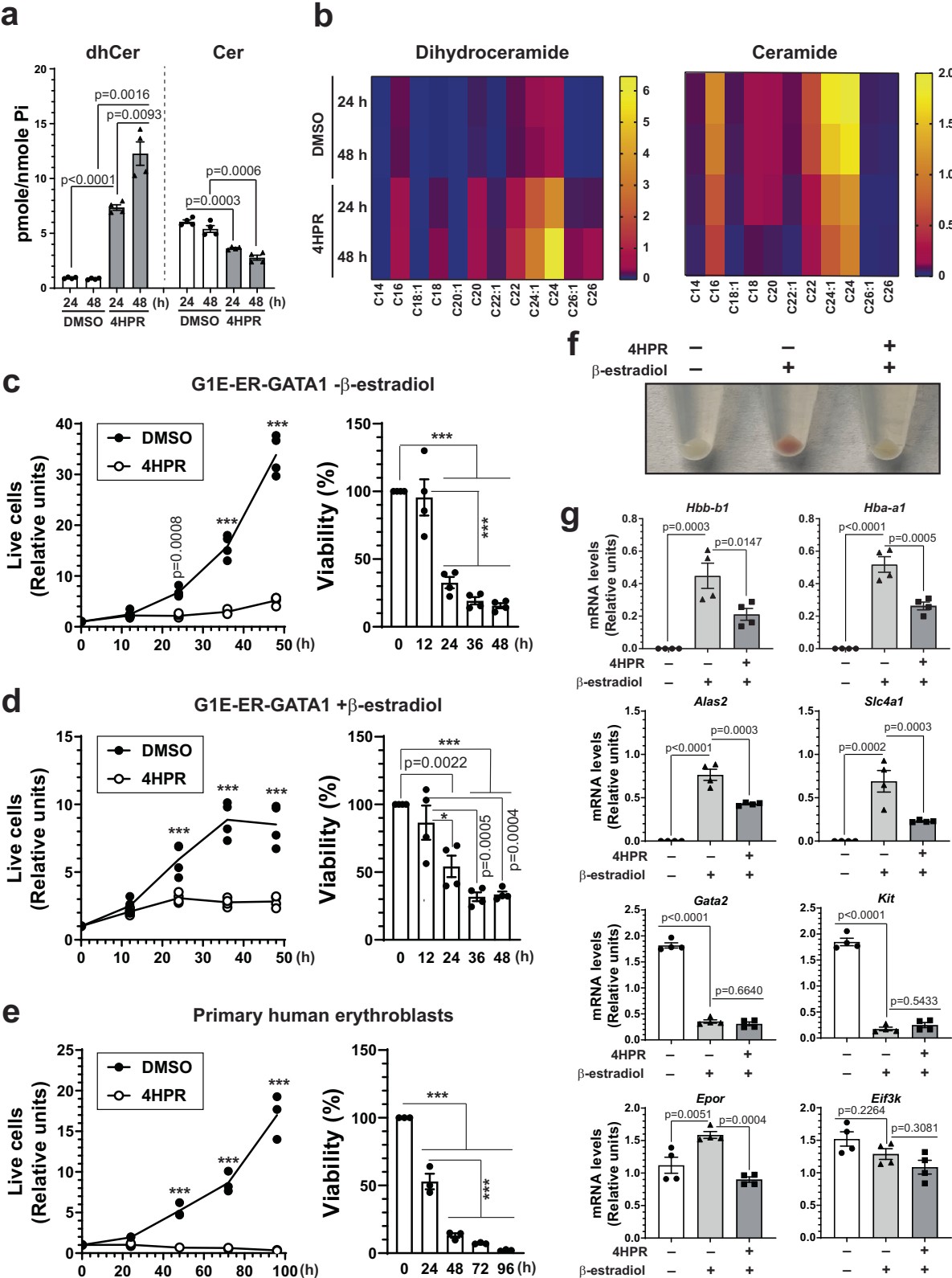

and reduced their steady-state levels (Fig. 5a and b). Pharmacologically inhibiting PP2A with okadaic acid (OA) rescued 4HPR and (dh)Cer-dependent reduction of p-AKT and p-ERK (Fig. 5a and b, Supplementary Fig. 5a and b), suggesting that disrupting Cer homeostasis inhibits cytokine signaling via PP2A. As OA abrogated p-STAT5, a potential role of PP2A in Cer-dependent inhibition of p-STAT5 could not be ascertained.

SET is an endogenous PP2A inhibitor that binds the catalytic C subunit of PP2A heterotrimer and inhibits its activity[38] (Fig. 5c). SET loss would be predicted to elevate PP2A activity. To genetically dissect how PP2A links Cer homeostasis with cytokine signaling, we used CRISPR-Cas9 to generate *Set*[-/-] G1E-ER-GATA1 cells. Western blotting revealed the quantitative depletion of SET in Set[-/-] cells (Fig. 5d). Under these conditions, Epo-induced p-AKT, p-ERK, and p-STAT5 were

**Fig. 2 | Ceramide homeostasis is required for erythroblast survival and differentiation. a** G1E-ER-GATA1 cells were treated with DMSO or 2 μM 4HPR for 24 or 48 h and Cer and dhCer levels were quantified by lipidomics (n = 4 biologically independent samples, mean ± SEM). The levels of lipids were normalized to the level of inorganic phosphate (Pi). p values were calculated by two-tailed paired Student's test. **b** Heatmaps depicting levels of individual species of Cer and dhCer from (a). **c, d** G1E-ER-GATA1 cells were treated with DMSO or 4HPR in the absence or presence of β-estradiol. Left, live cell numbers were quantified at 0, 12, 24, 36, and 48 h after treatment by Trypan Blue dye exclusion assay (n = 4 biologically independent samples). p values were calculated using two-way ANOVA followed by Sidak's multiple comparisons test. Right, viability was calculated by normalizing live cell numbers of 4HPR-treated samples to DMSO-treated ones at each time point (n = 4 biologically independent samples, mean ± SEM). p values were calculated using one-way ANOVA followed by Tukey's multiple comparisons test. *p = 0.0319, ***p < 0.0001. **e** Primary human G-CSF-mobilized mononuclear cells were differentiated towards the erythroid lineage. At D9, cells were treated with DMSO or 4HPR. Left, live cell numbers were quantified at 0, 24, 48, 72, and 96 h after treatment by Trypan Blue dye exclusion assay (n = 3 biologically independent samples). p values were calculated using two-way ANOVA followed by Sidak's multiple comparisons test. Right, viability was calculated by normalizing live cell numbers of 4HPR-treated samples to DMSO-treated ones at each time point (n = 3 biologically independent samples, mean ± SEM). p values were calculated using one-way ANOVA followed by Tukey's multiple comparisons test. ***p < 0.0001. **f** Representative picture showing impaired hemoglobinization of 4HPR-treated cells after differentiation. **g** G1E-ER-GATA1 cells were treated with β-estradiol with or without 4HPR, and mRNA levels were quantified by RT-qPCR (n = 4 biologically independent samples, mean ± SEM). p values were calculated using one-way ANOVA followed by Dunnett's multiple comparisons test. Source data are provided as a Source Data file.

reduced in comparison with WT cells, while SCF-induced signaling was unaffected (Fig. 5e and f). Alternatively, we compared the activities of mechanistically distinct PP2A activators to C6-Cer. FTY720 binds SET, dissociating it from the catalytic subunit[39]. By contrast, DT-061 binds the interface of the PP2A A, B, and C subunits, stabilizing the heterotrimer[40] (Fig. 5c). Both PP2A activators resembled C6-Cer in attenuating SCF- and Epo-induced p-AKT, p-ERK, and p-STAT5 (Figs. 5g and h). Thus, PP2A activation suffices to repress SCF and Epo signaling pathways.

Since ceramides can activate PP2A by dissociating SET from the PP2A heterotrimer[41] (Fig. 5c), we asked if depleting SET alters the sensitivity of cytokine signaling to ceramide. C6-Cer dose-dependently reduced Epo-induced p-AKT, p-ERK, and p-STAT5 in WT and Set[-/-] cells (Fig. 5i). Although Epo-induced signaling was attenuated in Set[-/-] cells, the concentration-dependence for the Cer-dependent reduction, relative to the highest inducible signal, was identical in WT and Set[-/-] cells (Fig. 5j). Cer activity to reduce Epo signaling is therefore SET-independent and PP2A-dependent.

To extend mechanistic insights regarding how the GATA1-instigated network governs ceramide homeostasis, we investigated other GATA1-regulated components of the ceramide metabolic system. *Ormdl* is a family of three genes encoding proteins that bind serine palmitoyltransferase (SPT), suppressing the activity of this rate-limiting enzyme in sphingolipid biosynthesis[42] (Fig. 5k). The collective genetic ablation of all *ORMDL* genes in a lung cancer cell line strongly elevated endogenous sphingolipid levels[43]. In G1E-ER-GATA1 cells, GATA1 induced *Ormdl3* RNA expression 16-fold (p < 0.0001), while *Ormdl1* and *Ormdl2* expression were reduced by 33% (p = 0.0001) and 47% (p < 0.0001), respectively (Fig. 5l). GATA1 induced chromatin accessibility near the 5′- and 3′-ends of murine *Ormdl3*, at sites that are GATA1-occupied in G1E-ER-GATA1 and primary murine erythroblasts. In human PBDE, GATA1 occupied sites flanking the *ORMDL3* locus. The GATA1-occupied sequences contain WGATAR motifs, suggesting that *Ormdl3* is a GATA1 target gene (Supplementary Fig. 5c). Reducing Ormdl3 levels would be predicted to elevate serine palmitoyltransferase activity, disrupt Cer homeostasis, and attenuate cytokine signaling (Fig. 5k). We used CRISPR-Cas9 to generate *Ormdl3* knockout G1E-ER-GATA1 clonal cell lines (Fig. 5m). Western blotting with an antibody that reacts with multiple ORMDL proteins revealed their reduction in *Ormdl3[-/-]* cells in comparison to WT cells upon erythroid differentiation (Fig. 5n). Quantification of Epo signaling in WT and *Ormdl3[-/-]* cells 48 h post-differentiation revealed that Epo-induced p-AKT was reduced in *Ormdl3[-/-]* cells, while p-ERK1/2 and p-STAT5 were unaffected (Figs. 5o and 5p). Thus, genetically rewiring the Cer metabolic system by eliminating a GATA1-induced suppressor of sphingolipid biosynthesis, which is predicted to disrupt Cer homeostasis by elevating endogenous dhCer and Cer, was detrimental to Epo signaling. This analysis established Ormdl3 as a new component of a critical cytokine signaling mechanism. Using an alternative approach, we treated the cells with bacterial sphingomyelinase (bSMase) which increases endogenous Cer by hydrolyzing sphingomyelin at the plasma membrane, and measured SCF and Epo signaling. A 30 min bSMase treatment significantly reduced SCF-induced p-AKT and Epo-induced p-AKT and p-STAT5 (Supplementary Fig. 5d and e). Thus, disrupting ceramide homeostasis by manipulating endogenous Cer levels recapitulated concepts developed by our pharmacological and genetic analyses.

## Dual-component mechanism mediates ceramide-dependent suppression of cytokine signaling

Genetic manipulation of PP2A activity reduced Epo, but not SCF, signaling (Fig. 5), suggesting that Cer and/or PP2A differentially impact Epo and SCF signaling. To compare the sensitivities of Epo and SCF signaling to PP2A or Cer, we treated G1E-ER-GATA1 cells with increasing amounts of the PP2A activator DT-061 or C6-Cer and quantified Epo and SCF signaling by Western blotting. DT-061 caused a dose-dependent reduction of Epo- and SCF-induced p-AKT and p-ERK. Epo signaling was reduced greater than SCF signaling at sub-maximal doses (Fig. 6a and b). Similar results were detected with C6-Cer (Fig. 6c and d), suggesting that Epo signaling is disproportionately sensitive to Cer and PP2A activation. As Epo receptor activation requires JAK2 activity, we asked if PP2A and/or Cer antagonize JAK2. Both DT-061 and C6-Cer reduced Epo-induced JAK2 phosphorylation in a dose-dependent manner (Fig. 6e), suggesting that the Cer/PP2A mechanism opposes Epo-induced JAK2 activation. Inhibiting JAK2 with ruxolitinib (Rux) abrogated Epo-induced p-AKT, p-ERK, and p-STAT5, without affecting SCF signaling (Fig. 6f), suggesting that JAK2 inhibition contributes to the hypersensitivity of Epo signaling to Cer/PP2A.

To further dissect how establishing ceramide homeostasis commissions cytokine signaling, we asked if subjecting cells to an acute (30 min) Cer exposure would reveal the primary steps in Epo and SCF signaling that are Cer-sensitive and differentially sensitive. A 30 min C6-Cer treatment in G1E-ER-GATA1 cells reduced Epo or SCF-induced p-AKT and p-ERK without affecting Epo-induced p-STAT5 (Fig. 6g and h). Treatment of primary human erythroblasts for 30 min with C6-Cer reduced SCF and Epo-induced p-AKT, while Epo-induced p-STAT5 remained unchanged (Fig. 6i and j), indicating a differential sensitivity of signaling pathways to an acute elevation of Cer.

To determine if the Cer sensitivity of Epo and SCF signaling systems reflects inhibition of one pathway, which debilitates the other pathway, we tested if Epo- and SCF-induced AKT and ERK signaling pathways are independent or intertwined. The cells were treated with MEK (U0126) or PI3K (wortmannin) inhibitors, and SCF and Epo signaling was quantified by Western blotting. U0126 abrogated Epo and SCF-induced p-ERK, while p-AKT was unaffected. By contrast, wortmannin abrogated SCF- and Epo-induced p-AKT, while p-ERK was unaffected (Supplementary Fig. 6a). Neither treatment impacted Epo-induced p-STAT5, suggesting that the pathways operate independently of each other.

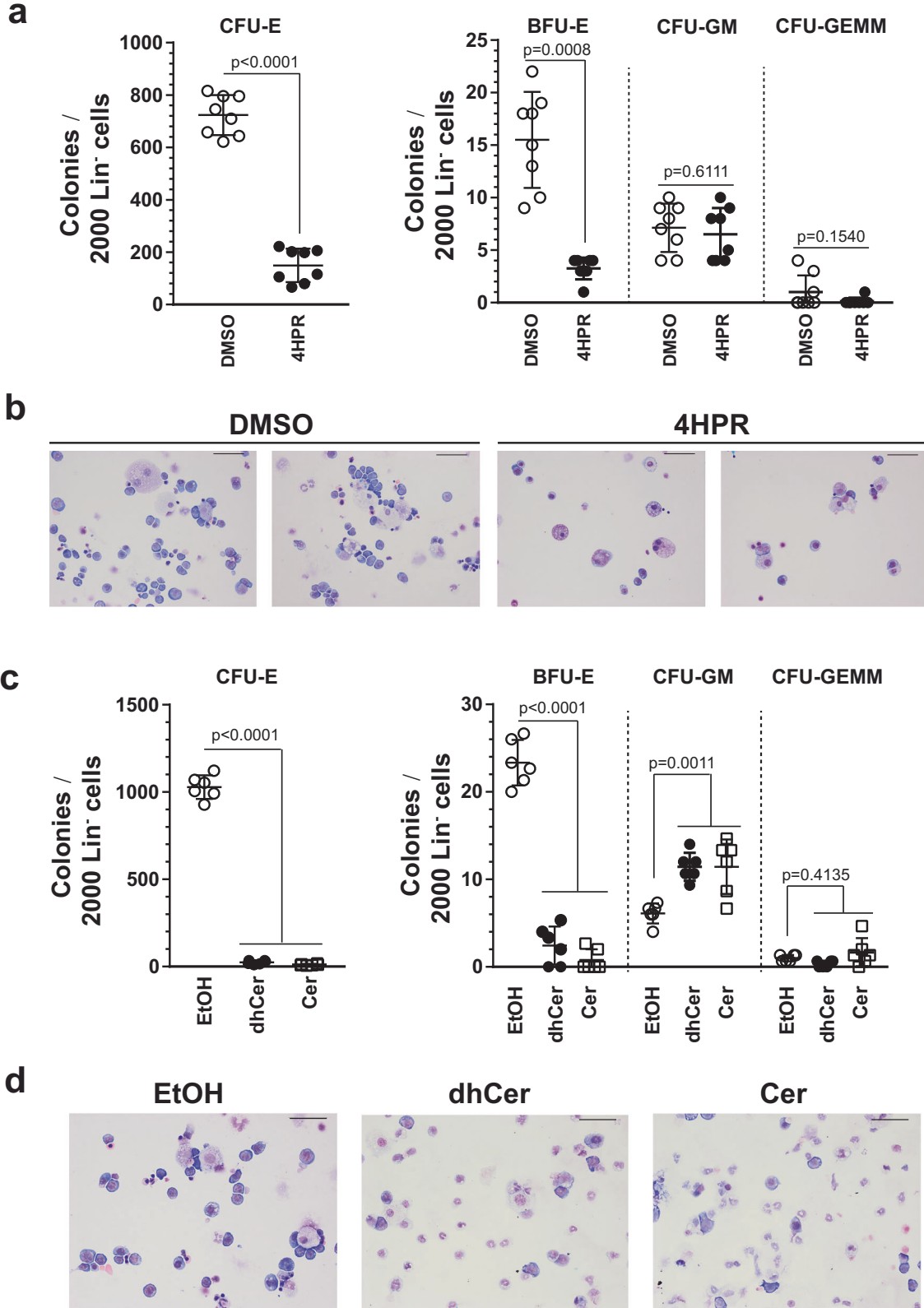

**Fig. 3 | Disrupting ceramide homeostasis is detrimental to erythroid, but not myeloid, progenitor function. a** CFU activity of Lin⁻ cells from E14.5 fetal livers treated with or without 4HPR (*n* = 8 biologically independent samples from 2 independent experiments, mean ± SD). *p* values were calculated by two-tailed unpaired Student's test. **b** Representative Wright-Giemsa staining of cells recovered from day 8 of CFU assays in (**a**). Scale bars = 50 µm. **c** CFU activity of Lin⁻ cells from E14.5 fetal livers treated with vehicle (EtOH), C6-dhCer, or C6-Cer (*n* = 6 biologically independent samples from 2 independent experiments, mean ± SD). *p* values were calculated using one-way ANOVA followed by Dunnett's multiple comparisons test. **d** Representative Wright-Giemsa staining of cells recovered from day 8 of CFU assays in (**c**). Scale bars = 50 µm. Source data are provided as a Source Data file.

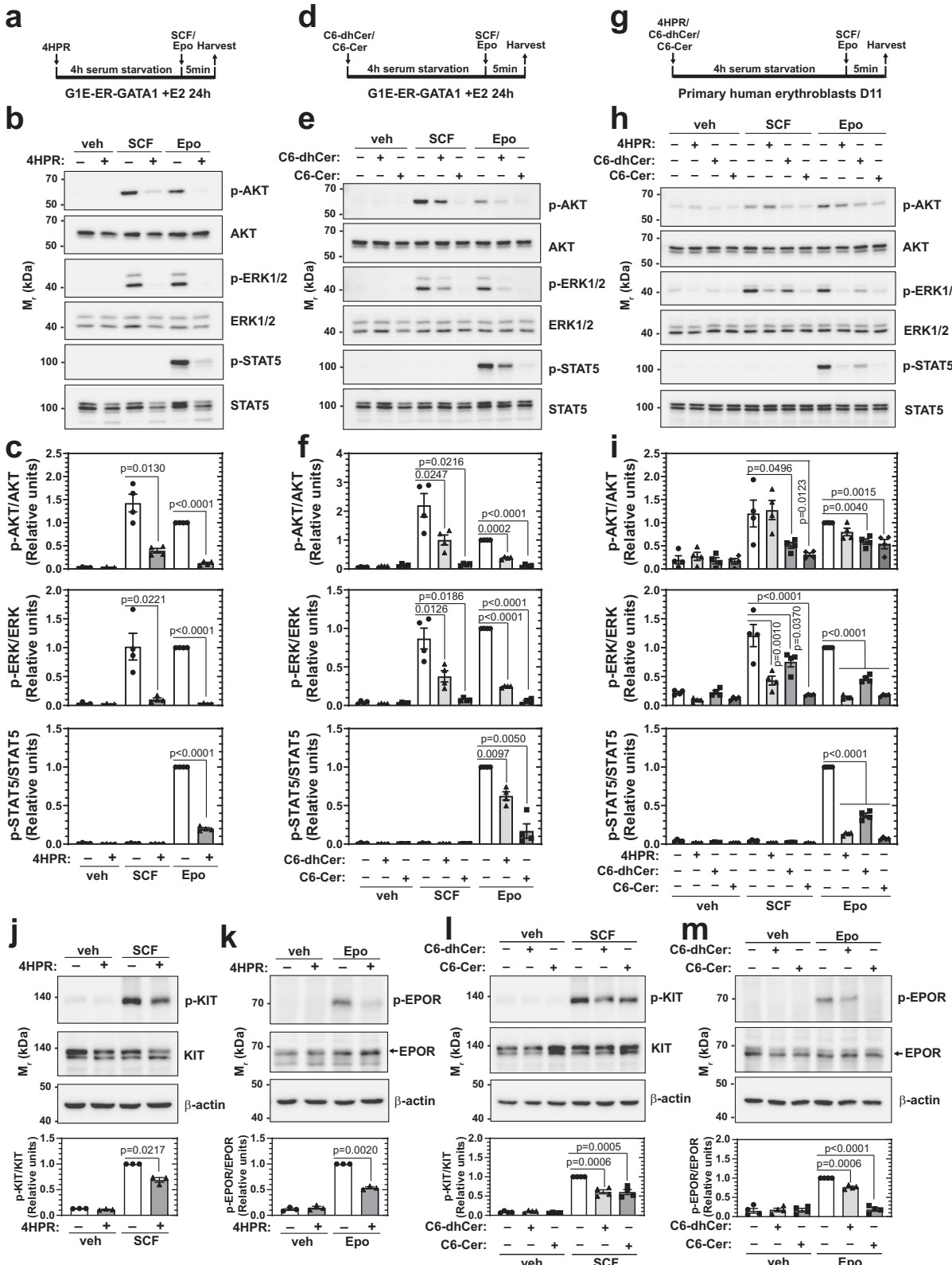

The independence of the Cer-sensitive pathways suggested that ceramides may oppose cytokine signaling by antagonizing ligand-dependent activation of the Epo and SCF receptors or activities of downstream signaling components utilized by both receptors. A 30 min C6-Cer treatment reduced SCF-induced p-AKT without affecting SCF-induced phosphorylation of its receptor c-KIT (Fig. 6k and l). Thus, the acute ceramide treatment inhibited SCF/c-KIT signaling via a post-receptor mechanism. The acute ceramide treatment also reduced FBS-induced p-AKT and p-ERK (Supplementary Fig. 6b and c), suggesting that AKT and ERK pathways are inhibited by Cer regardless of the receptor system. By contrast, acute ceramide treatment reduced Epo-dependent induction of p-JAK2 and p-EPOR (Fig. 6m and n), suggesting that cytokine-dependent activation of EPOR is more sensitive to Cer homeostasis than KIT. These results support a model in which

**Fig. 4 | Ceramide homeostasis commissions cytokine signaling in human and murine erythroblasts. a–c** G1E-ER-GATA1 cells were treated with β-estradiol (E2) for 24 h, serum-starved, treated with 2 μM 4HPR for 4 h, and stimulated with SCF or Epo for 5 min. SCF and Epo signaling were measured by Western blotting (**b**). Quantitation of p-AKT, p-ERK, and p-STAT5 is shown in (**c**). Values were normalized to their respective protein levels (*n* = 4 independent experiments, mean ± SEM). *p* values were calculated by two-tailed paired Student's test. **d–f** G1E-ER-GATA1 cells were treated with β-estradiol for 24 h, serum-starved, treated with 25 μM C6-dhCer or C6-Cer for 4 h, and stimulated with SCF or Epo for 5 min. SCF and Epo signaling were measured by Western blotting (**e**). Quantitation of p-AKT, p-ERK, and p-STAT5 signals is shown in (**f**). Values were normalized to their corresponding protein levels (*n* = 4 independent experiments, mean ± SEM). *p* values were calculated using one-way ANOVA followed by Dunnett's multiple comparisons test. **g–i** Primary human erythroblasts at D11 were serum-starved, treated with 2 μM 4HPR, 25 μM C6-dhCer, or 25 μM C6-Cer for 4 h, and stimulated with SCF or Epo for 5 min. SCF and Epo signaling were measured by Western blotting (**h**). Quantitation of p-AKT, p-ERK, and p-STAT5 signals is shown in (**i**). Values were normalized to their corresponding protein levels (*n* = 4 independent experiments, mean ± SEM). *p* values were calculated using one-way ANOVA followed by Dunnett's multiple comparisons test. **j, k** G1E-ER-GATA1 cells were treated with 4HPR for 4 h and cytokine-dependent phosphorylation of receptors (KIT and EPOR) were measured by Western blotting. Values were normalized to their corresponding protein levels (*n* = 3 biologically independent samples, mean ± SEM). *p* values were calculated by two-tailed paired Student's test. **l, m** G1E-ER-GATA1 cells were treated with C6-dhCer or C6-Cer for 4 h and cytokine-dependent phosphorylation of receptors (KIT and EPOR) were measured by Western blotting. Values were normalized to their corresponding protein levels (*n* = 4 biologically independent samples, mean ± SEM). *p* values were calculated using one-way ANOVA followed by Dunnett's multiple comparisons test. Source data are provided as a Source Data file.

disrupting Cer homeostasis impairs cytokine signaling through a dual-component mechanism. Acutely disrupting Cer homeostasis compromised AKT and ERK signaling systems post-receptor activation, and Epo-dependent JAK2 and EPOR activation were hypersensitive to these alterations (Fig. 6o).

## Discussion

By amalgamating multiomic datasets in a genetic complementation system, we discovered that GATA1 regulates genes encoding sphingolipid metabolic enzymes, thereby controlling Cer homeostasis during erythroid differentiation. As GATA1 is vital for conferring erythroblast survival and differentiation[44–48], we reasoned that the downstream signaling lipids might mediate important GATA1 functions. Using orthogonal approaches, we demonstrated that disrupting Cer homeostasis impairs cytokine signaling pathways that are essential for erythroid cell survival, proliferation, and differentiation (Fig. 7). These results establish that Cer controls one of the most fundamental systems required for the development of the erythrocyte – the core signal transduction machinery. Bioactive lipids are integral components of GATA factor-dependent regulatory networks in mouse and human systems.

GATA1 regulates the expression of *Degs1*, which confers Cer homeostasis by converting dhCer to Cer. By contrast to Cer, which controls multiple cellular processes[47–49], dhCer was considered to be inactive. However, recent studies indicate that dhCer has activity to regulate apoptosis, proliferation, and autophagy[8,50]. γ-Tocopherol treatment, which increased dhCer levels, induced apoptosis in prostate cancer cells[51]. siRNA knockdown of *DEGS1* or pharmacologically inhibiting DES with 4HPR, which increases dhCer, caused cell cycle arrest at G0/G1, and reduced Rb phosphorylation in human neuroblastoma cells[26]. 4HPR-mediated DES inhibition enhanced self-renewal of human hematopoietic stem cells ex vivo[9]. Disrupting Cer homeostasis by inhibiting DES or elevating short chain dhCer inhibited erythroid, but not myeloid, progenitor functions, providing evidence for a cell-type specific action of dhCer. Our results revealed that, in certain contexts, dhCer has activities similar to Cer to control cell proliferation and/or survival. The analyses involving DES inhibition with 4HPR, C6-dhCer and C6-Cer, sphingomyelinase, and genetic manipulation of sphingolipid biosynthesis, indicate that both elevated levels of dhCer and Cer deleteriously impact erythroid, but not myeloid, progenitor function and cytokine signaling. In most cases, Cer was more potent and effective. As sphingolipid biosynthesis increases during erythroid development to produce Sph-1P that is released from erythrocytes, GATA1-dependent regulation of sphingolipid metabolic enzymes and *Ormdl3* confer ceramide homeostasis by preventing excessive accumulation of dhCer or Cer, and ceramide homeostasis commissions SCF and Epo signaling. Our lipidomic analysis revealed a ~30% increase in Cer during erythroid differentiation. However, our study did not distinguish between models in which the increased Cer

exclusively reflects an intermediate for Sph-1P production or exerts functional activity to control differentiation.

SCF and Epo signaling systems are vital for erythropoiesis[52–60]. In addition, SCF has broader roles to regulate hematopoietic stem and progenitor cells, and non-erythroid Epo actions have been described. While SCF/c-Kit confers survival and proliferation of hematopoietic stem and progenitor cells, including erythroid progenitors[61], Epo/EpoR promotes survival of committed erythroid progenitors (CFU-E) and terminal differentiation[59]. In certain contexts, SCF and Epo may function collectively[32]. SCF binding to the c-Kit receptor tyrosine kinase triggers c-Kit dimerization and autophosphorylation, which recruits signaling adaptors and activates PI3K/AKT and Ras/Raf/MAPK pathways[61]. By contrast, the Epo receptor lacks intrinsic tyrosine kinase activity and relies on Janus kinase 2 (JAK2) recruitment for its signal transduction[62]. Epo binding activates JAK2, which phosphorylates EpoR at multiple cytoplasmic tyrosines and recruits Src-homology 2 (SH2) domain-containing proteins, including STAT5, p85-PI3K (which activates AKT), and Grb2 (which activates ERK)[31,63–65]. Our studies demonstrated that establishing Cer homeostasis commissions SCF and Epo signaling. Although disrupting Cer homeostasis with an acute Cer exposure inhibited AKT and ERK phosphorylation post-receptor activation, it also inhibited Epo-induced JAK2 and EpoR activation, which impaired Epo-induced p-AKT, p-ERK, and p-STAT5. This dual-component inhibitory mechanism explains the erythroid cell hypersensitivity to disrupted Cer homeostasis.

Biochemical studies utilizing purified PP2A subunits revealed that exogenous short-chain Cer enhances PP2A heterotrimer activity, while endogenous C18-ceramide enhances the activity of the PP2A catalytic subunit and the PP2A heterotrimer[37]. Activation is stereospecific and regulated by phosphatidic acid[66]. Cer also binds directly to the endogenous PP2A inhibitor SET, dissociating it from PP2A, indirectly enhancing PP2A activity[41]. As targeted ablation of *Set* did not alter the sensitivity of SCF and Epo signaling to Cer, the Cer-dependent mechanism to inhibit cytokine signaling is SET-independent. Although it is reasonable to assume that the inhibitory Cer activity involves direct activation of PP2A, one cannot rule out the possibility that Cer enhances PP2A activity via other endogenous inhibitors, *e.g.*, CIP2A and PME-1[67]. As a Cer/PP2A circuit inhibits insulin and β-adrenergic receptor signaling in liver and adipose cells[68–70], the mechanism elucidated herein might be extrapolated to other receptor systems.

In summary, we demonstrated that a GATA1-dependent transcriptional mechanism establishes Cer homeostasis to commission cytokine signaling in erythroid cells via a PP2A-dependent, dual-component mechanism. Integrating signaling lipids into GATA factor-dependent regulatory networks provides a new dimension on how GATA factors control cell state transitions and is expected to have broader implications in diverse systems.

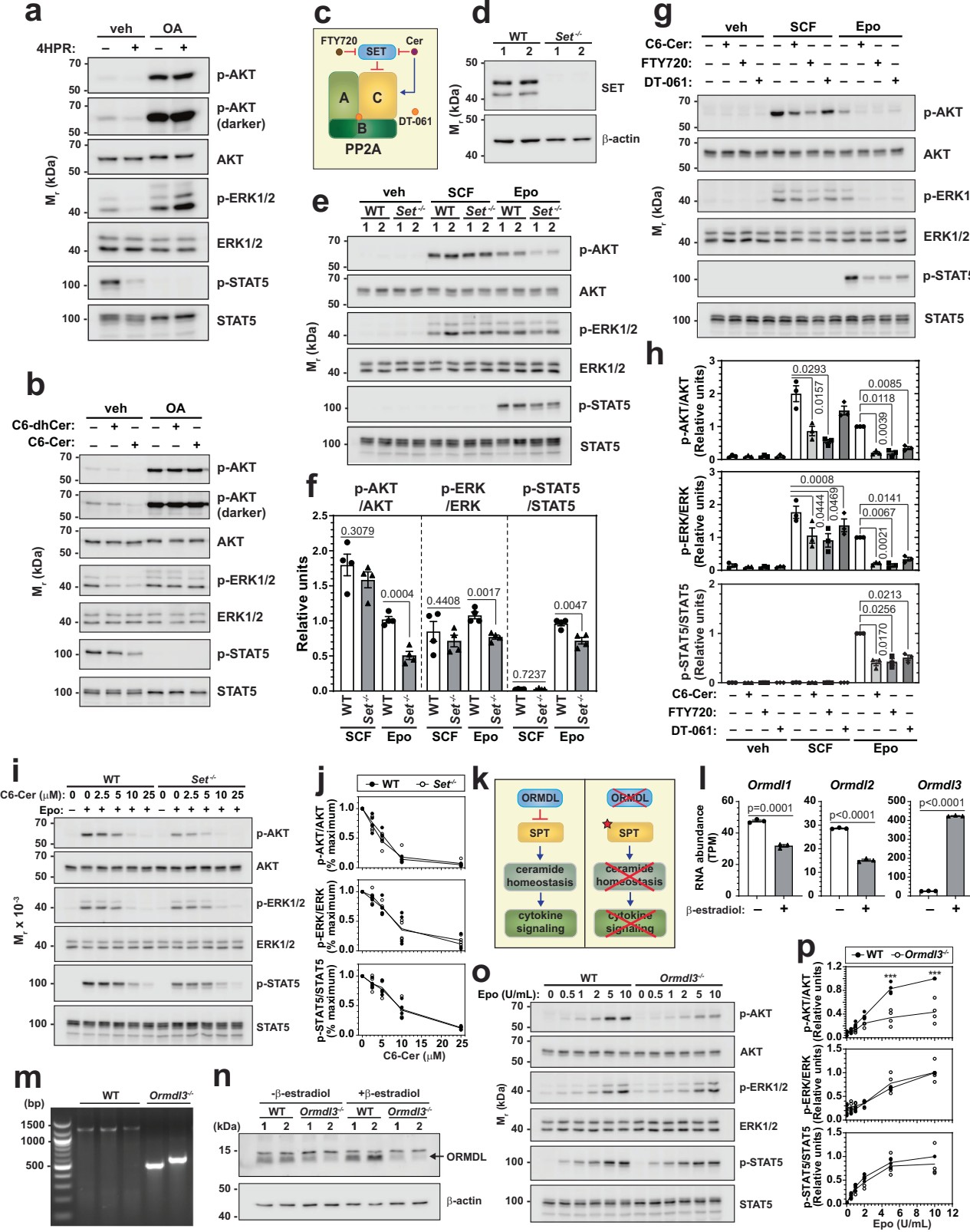

## Methods

### Mice

C57BL/6 J mice (The Jackson Laboratory) were used. Mice were housed in a facility with 12 h/12 h light/dark cycle, ambient temperature at 20–25°C, and humidity at 40-60%. All animals were handled according to approved institutional animal care and use committee (IACUC) protocols (#M02230) of the University of Wisconsin-Madison. All

animal experiments were performed with the ethical approval of the AAALAC International (Association for the Assessment and Accreditation of Laboratory Animal Care) at the University of Wisconsin-Madison.

### Cell lines and primary cultures

G1E-ER-GATA1 cells[14] derived from GATA1-mutant male murine ES cells[71] were cultured in Iscove's modified Dulbecco's medium (IMDM;

**Fig. 5 | Genetic rewiring of ceramide-cytokine signaling circuit revealed a PP2A-dependent, multi-component mechanism. a, b** G1E-ER-GATA1 cells were pre-treated with vehicle or 100 nM okadaic acid (OA) for 1 h followed by 4HPR, C6-dhCer, or C6-Cer treatment for 3 h. Western blotting was conducted to measure p-AKT, p-ERK1/2, and p-STAT5 levels ($n$ = 3 biologically independent samples). Quantitation of blots were shown in Supplementary Fig. 5a and b. **c** Diagram illustrating the mechanism of SET, Cer, FTY720, and DT-061 to modulate PP2A activity. **d** CRISPR-Cas9 was used to generate *Set*[-/-] G1E-ER-GATA1 clones. Representative Western blots from 3 independent experiments demonstrated the quantitative depletion of SET protein in two *Set*[-/-] clones. **e** SCF and Epo signaling in two WT and two *Set*[-/-] clonal lines were measured by Western blotting. **f** Quantitation of blots in (**e**). p-AKT, p-ERK, and p-STAT5 signals were normalized to their corresponding protein levels ($n$ = 4 biologically independent samples, mean ± SEM). *p* values were calculated by two-tailed unpaired Student's test. **g** G1E-ER-GATA1 cells were serum starved, treated with 10 μM C6-Cer, FTY720, or DT-061 for 4 h, and stimulated with SCF or Epo. Western blotting was conducted to measure cytokine signaling. **h** Quantitation of blots in (**g**). p-AKT, p-ERK, and p-STAT5 signals were normalized to their respective protein levels ($n$ = 3 independent experiments, mean ± SEM). *p* values were calculated using one-way ANOVA followed by Dunnett's multiple comparisons test. **i** WT or *Set*[-/-] cells were treated with increasing concentration of C6-Cer before stimulation with Epo. Two WT and two

*Set*[-/-] clones were used. Western blotting was conducted to measure Epo signaling. **j** Quantitation of blots in (**i**). Ceramide-dependent inhibition of signaling was assessed by normalizing phosphorylation signals to the maximally induced signal in each cell type ($n$ = 4 independent experiments). Statistical analyses were performed using two-way ANOVA followed by Sidak's multiple comparisons test, suggesting no significant difference between WT and *Set*[-/-] cells. **k** Diagram illustrating ORMDL-dependent regulation of ceramide homeostasis and cytokine signaling. **l** GATA1-dependent regulation of *Ormdl1*, *2*, and *3* mRNA in G1E-ER-GATA1 cells ($n$ = 3 biologically independent samples, mean ± SEM). *p* values were calculated by two-tailed unpaired Student's test. **m** *Ormdl3*[-/-] clones were generated by CRISPR-Cas9. Representative gel of PCR products from 3 independent experiments confirmed the homozygous deletion at *Ormdl3* locus. **n** Representative Western blots from 3 independent experiments showing that ORMDL proteins were quantitatively reduced in two *Ormdl3*[-/-] clones after erythroid differentiation. **o** WT and *Ormdl3*[-/-] cells were serum-starved and stimulated with Epo at increasing concentrations. Two WT and two *Ormdl3*[-/-] clonal lines were used. Epo signaling was measured by Western blotting. **p** Quantitation of blots in (**o**). Phosphorylation signals were normalized to their corresponding protein levels ($n$ = 4 independent experiments). *p* values were calculated using two-way ANOVA followed by Sidak's multiple comparisons test. ***$p$ < 0.0001. Source data are provided as a Source Data file.

Gibco) containing 15% FBS (Gemini), 1% penicillin-streptomycin (Gemini), 2 U/ml erythropoietin (Amgen), 120 nM monothioglycerol (Sigma), and 0.6% conditioned medium from a Kit ligand (SCF)-producing CHO cell line, and 1 μg/ml puromycin (Gemini). ER-GATA1 activity was induced by adding 1 μM β-estradiol (Steraloids) to the media. hi-WT cells[29] were cultured in OPTI-MEM supplemented with 10% FBS (Gemini), 1% penicillin-streptomycin (Gemini), 1% SCF-conditioned medium, 30 μM β-mercaptoethanol (Sigma), 1 μM β-estradiol (Steraloids), and 500 μg/ml G418 (Gemini).

Primary human mononuclear cells were isolated from G-CSF-mobilized peripheral blood using Histopaque (Sigma). The sex of the primary cells used for experiments is not available. Cells were maintained in StemSpan SFEM medium (Stem Cell Technologies) supplemented with 1x CC100 cytokine mix (Stem Cell Technologies) for 4 days. To induce differentiation, cells were cultured in basal differentiation media [IMDM containing 15% FBS, 2 mM L-glutamine, 1% BSA (Sigma), 500 μg/ml holo human transferrin (Sigma), and 10 μg/ml insulin (Sigma)] supplemented with 1 μM dexamethasone (Sigma), 1 μM β-estradiol (Steraloids), 5 ng/ml IL-3 (Peprotech), 100 ng/ml SCF (Peprotech), and 6 U/ml erythropoietin (Amgen) for 5 days, followed by 4 days in basal differentiation media supplemented with 50 ng/ml SCF and 6 U/ml erythropoietin. Cells were cultured in a humidified 5% $CO_2$ incubator at 37 °C.

For cytokine signaling experiments, cells were transferred to serum-free media (IMDM or IMDM + 1% BSA) and incubated at 37 °C for 4 h before stimulating with 100 ng/ml SCF (Peprotech) or 10 U/ml erythropoietin (Amgen) for 5 min. To disrupt ceramide homeostasis, cells were treated with 4HPR (TCI Chemicals), C6-dhCer (Sigma), or C6-Cer (Sigma) for 4 h or 30 min before cytokine stimulation. To inhibit cytokine signaling pathways, cells were treated with wortmannin (Cell Signaling Technology), U0126 (Cell Signaling Technology), or ruxolitinib (Selleckchem) for 1 h before cytokine stimulation. For PP2A activation, cells were treated with FTY-720 (Sigma) or DT-061 (Selleckchem) for 4 h before cytokine stimulation. For bSMase experiments, cells were treated with SMase from *Bacillus cereus* (Sigma) at 0.5 U/ml for 30 min before cytokine stimulation.

All cell-based studies were conducted according to approved biosafety protocol B00000056-AM008 at University of Wisconsin-Madison.

## RT-qPCR

Total RNA was purified from 0.5 to 2 × 10⁶ cells with TRIzol (Life Technologies). 1 μg RNA was treated with DNase I (Life Technologies)

for 15 min at room temperature. DNase I was inactivated by addition of EDTA and heating at 65 °C for 10 min. To synthesize cDNA, DNase I-treated RNA was incubated with 125 ng of a 5:1 mixture of oligo-dT primers and random hexamer at 68 °C for 10 min. RNA/primers were incubated with Moloney MLV reverse transcriptase (Life Technologies), 10 mM DTT, RNAsin (Promega), and 0.5 mM deoxynucleoside triphosphates (dNTP) at 42 °C for 1 h, and then heat inactivated at 98 °C for 5 min. Real-time PCR (RT-qPCR) was conducted with Power SYBR Green Master Mix (Applied Biosystems) using ViiA 7 Real-Time PCR system and QuantStudio v1.6.1 (Applied Biosystems). Primers used are listed in Supplementary Table 1.

## Western blotting

1-2 × 10⁶ cells were boiled in SDS lysis buffer (50 mM Tris, pH 6.8, 2% β-mercaptoethanol, 2% SDS, 0.04% bromophenol blue, 10% glycerol) for 15 min. Samples were resolved by SDS–PAGE, transferred to PVDF membrane, blocked with 5% bovine serum albumin (BSA, Sigma), and incubated with primary antibody overnight. Antibodies against AKT, ERK1/2, JAK2, KIT, and p-GATA1 were diluted 1:2000. Antibodies against STAT5, GATA1, GATA2, and EPOR were diluted 1:5,000. Antibody against β-actin was diluted 1:10,000. All other antibodies were diluted 1:1,000. Membranes were washed in TBST, incubated with secondary antibody at 1:5000 (donkey antirabbit or donkey antimouse HRP-conjugated antibody, Jackson ImmunoResearch Laboratories), and detected with SuperSignal™ West Femto Maximum Sensitivity Substrate (Thermo Fisher Scientific). Blot images were collected and analyzed using ImageStudio or ImageStudio Lite v5.2 (LI-COR Biosciences). Sources for primary and secondary antibodies are listed in Supplementary Table 2.

## Gene editing with CRISPR-Cas9

Guide sequences targeting exons of murine *Set* and *Ormdl3* genes were designed using online tools (https://www.idtdna.com/).

*Set*_1: ACCATCCACAAGGTATGTGG
*Set*_2: ACATACTCACTGTGCTAAGC
*Ormdl3*_1: ACACGGGTGATGAACAGTCG
*Ormdl3*_2: ACTGGGAGCAGATGGACTAC

Chemically modified crRNAs and tracrRNA were purchased from IDT and Cas9 protein was purchased from Aldevron. 100 pmole of crRNA was mixed with 100 pmole of tracrRNA and incubated at 37 °C for 30 min. 8 μg of Cas9 protein was added to the mixture and incubated at 37 °C for 15 min to form RNP complex. 2 ×10⁵ G1E-ER-GATA1 cells were resuspended in 20 μl P3 buffer with 22% supplement (Lonza)

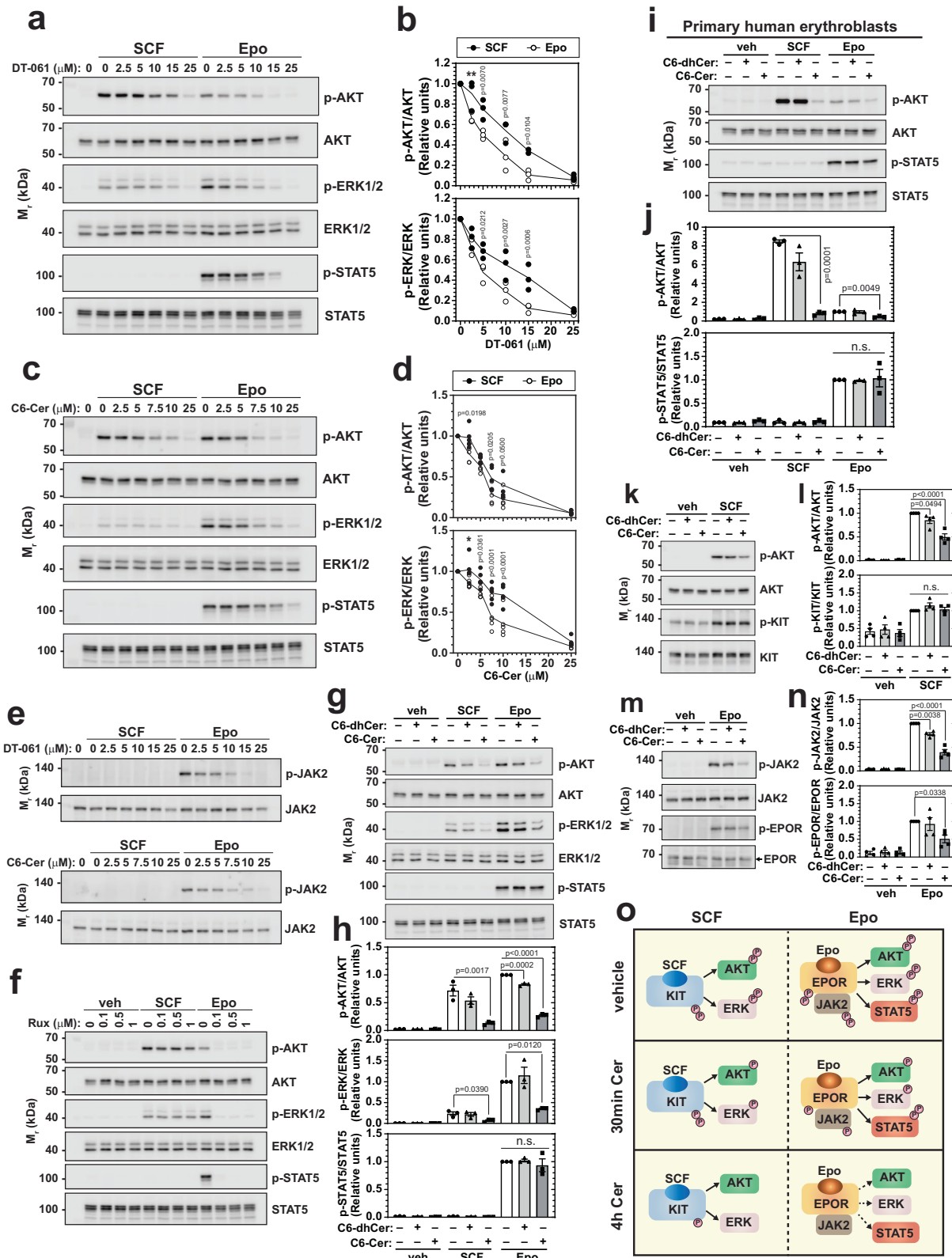

and added to the RNP complex. Electroporation was done using EO-100 program on Nucleofector 4D (Lonza). 72 h after nucleofection, cells were diluted in 96-well plates to isolate single-cell clones.

## Colony-forming unit assay
Fetal livers from E14.5 embryos were harvested on ice in PBS containing 2% FBS, 10 mM glucose, and 2.5 mM EDTA. Cells were

dissociated and passed through a single-cell strainer. Cells expressing lineage markers were removed using biotin-conjugated antibodies CD3e, CD11b, CD19, CD45R (B220), GR-1, Ter119, CD71, and MojoSort Streptavidin Nanobeads, all purchased from BioLegend. The remaining Lin- cells were plated in M3434 methylcellulose media (Stem Cell Technologies) at 2,000 cells per 35-mm dish with 2 µM 4HPR, 25 µM C6-dhCer, 25 µM C6-Cer, or equal volume of vehicle. CFU-Es were

**Fig. 6 | Ceramide/PP2A inhibits cytokine-induced AKT, ERK, and JAK/STAT pathways via distinct mechanisms. a** G1E-ER-GATA1 cells were treated with increasing concentrations of DT-061 for 4 h before stimulation with SCF or Epo. Western blotting was conducted to measure SCF and Epo signaling. **b** Quantitation of blots in (**a**). p-AKT and p-ERK signals were normalized to their respective protein levels and then normalized to the maximal induction by SCF and Epo, respectively ($n = 3$ independent experiments). $p$ values were calculated using two-way ANOVA followed by Sidak's multiple comparisons test. **$p = 0.0097$. **c** Cells were treated with increasing concentrations of C6-Cer for 4 h before stimulation with SCF or Epo. Western blotting was conducted to measure SCF and Epo signaling. **d** Quantitation of blots in (**c**). p-AKT and p-ERK signals were normalized to their respective protein levels and then normalized to the maximal induction by SCF and Epo, respectively ($n = 5$ independent experiments). $p$ values were calculated using two-way ANOVA followed by Sidak's multiple comparisons test. *$p = 0.0137$. **e** Cells were treated with increasing concentrations of DT-061 or C6-Cer for 4 h before stimulation with SCF or Epo. Western blotting was conducted to measure JAK2 phosphorylation ($n = 3$ independent experiments). **f** Cells were treated with rux-olitinib (Rux) at increasing concentrations before SCF or Epo treatment. Western blotting was conducted to measure SCF and Epo signaling ($n = 3$ independent experiments). **g** Cells were treated with 25 μM C6-dhCer or C6-Cer for 30 min before stimulation with SCF or Epo. Western blotting was conducted to measure SCF and Epo signaling. **h** Quantitation of blots in (**g**). Phosphorylation signals were

normalized to their respective protein levels ($n = 3$ independent experiments, mean ± SEM). $p$ values were calculated using one-way ANOVA followed by Dunnett's multiple comparisons test. n.s., not significant. **i** Primary human G-CSF-mobilized mononuclear cells were differentiated towards the erythroid lineage. D11 cells were serum-starved, treated with 25 μM C6-dhCer or C6-Cer for 30 min, and stimulated with SCF or Epo. Western blotting was conducted to measure SCF and Epo signaling. **j** Quantitation of blots in (**i**). Phosphorylation signals were normalized to their respective protein levels ($n = 3$ biologically independent samples, mean ± SEM). $p$ values were calculated using one-way ANOVA followed by Dunnett's multiple comparisons test. n.s., not significant. **k** Cells were treated with 10 μM C6-dhCer or C6-Cer for 30 min before stimulation with SCF. AKT and c-KIT phosphorylation was measured by Western blotting. **l** Quantitation of blots in (**k**). Phosphorylation signals were normalized to their respective protein levels ($n = 4$ biologically independent samples, mean ± SEM). $p$ values were calculated using one-way ANOVA followed by Dunnett's multiple comparisons test. n.s., not significant. **m** Cells were treated with 10 μM C6-dhCer or C6-Cer for 30 min before stimulation with Epo. JAK2 and EPOR phosphorylation was measured by Western blotting. **n** Quantitation of blots in (**m**). Phosphorylation signals were normalized to their respective protein levels ($n = 4$ biologically independent samples, mean ± SEM). $p$ values were calculated using one-way ANOVA followed by Dunnett's multiple comparisons test. **o** Model for signaling pathway disruption impacted by acute and prolonged ceramide treatment. Source data are provided as a Source Data file.

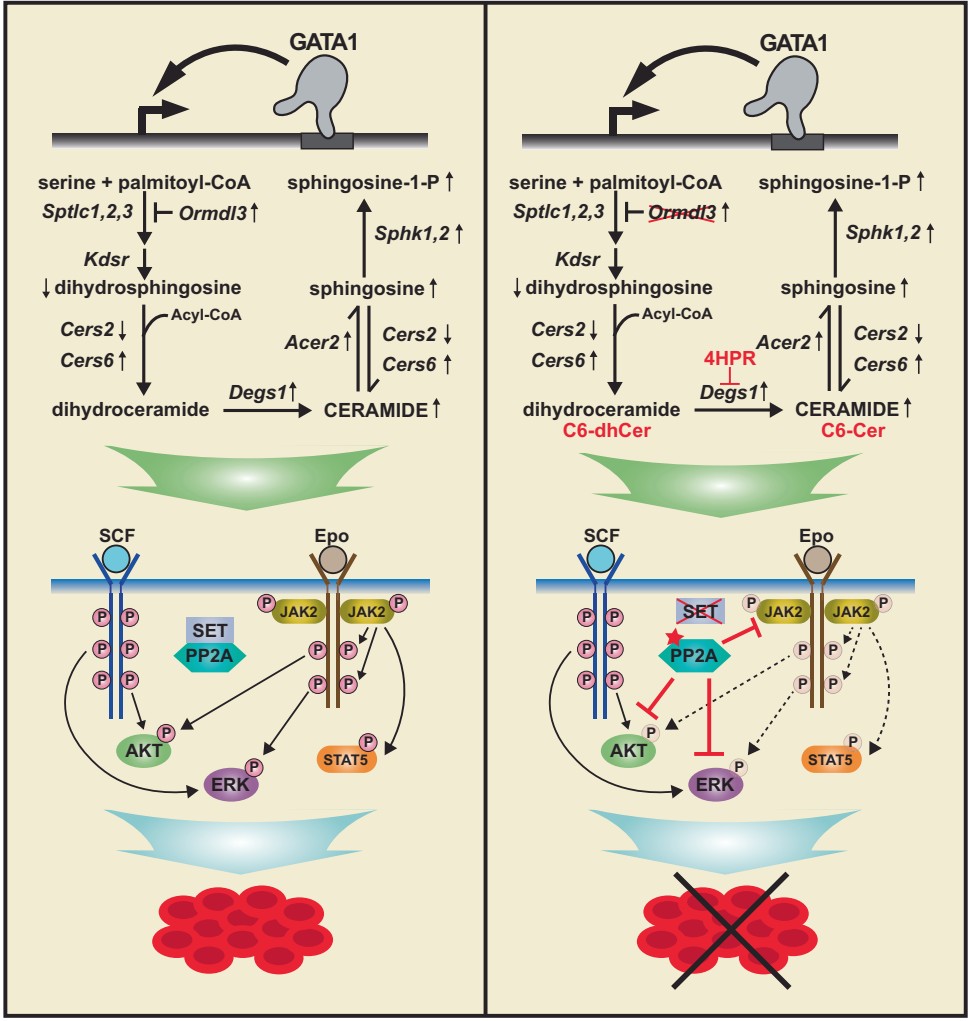

**Fig. 7 | Proposed model of GATA1 conferring ceramide homeostasis to commission cellular signaling mechanisms that govern erythropoiesis.** GATA1 controls ceramide homeostasis during erythroid development by regulating sphingolipid biosynthetic enzymes. Disrupting ceramide homeostasis deleteriously impacts erythropoiesis by antagonizing essential SCF and Epo signaling via PP2A-dependent mechanisms.

enumerated after 3 d. BFU-E, CFU-GM, and CFU-GEMM were enumerated after 8 d. Cells were recovered from day 8 cultures and subjected to Wright Giemsa staining (Sigma, WG16-500ML).

### Flow cytometry

For apoptosis analysis, cells were washed with PBS and then washed with Annexin V binding buffer (10 mM HEPES, 140 mM NaCl, 2.5 mM $CaCl_2$, pH 7.4). Cells were stained with Alexa Fluor® 647-Annexin V (BioLegend, 640912, 1:40) in 100 µl Annexin V binding buffer and incubated at RT for 15 min. Cells were then washed with 1 ml Annexin V binding buffer and resuspended in 500 µl Annexin V binding buffer + 1 µg/ml DAPI before analysis.

For erythroid differentiation assay, cells were washed with PBS + 2% FBS and stained with PE-CD71 (BioLegend, 113808, 1:100) and APC-Ter119 (BioLegend, 116212, 1:100) for 30 min on ice. Cells were then washed with 2% FBS, 10 mM glucose and 2.5 mM EDTA in PBS. All the above cells were analyzed on an Attune™ NxT Flow Cytometer (Thermo Fisher Scientific) and the data were analyzed using FlowJo v10.1.

### Quantitative lipidomics for sphingolipids

G1E-ER-GATA1 cells were treated with or without β-estradiol or 4HPR for 24 and 48 h and 2-4 ×$10^6$ cells were harvested, washed with PBS, and frozen as cell pellets. Lipid extraction and mass spectrometry for ceramide species, dihydroceramide species and sphingoid species using internal standards were performed by the Lipidomics Facility of Stony Brook University Medical Center.

Lipid extracts were mostly prepared as described by Bielawski et al.[72]. Approximately 3 ×$10^6$ cells were extracted with a total of 4 ml of 70% isopropanol:ethyl acetate (2:3,v/v) containing appropriate internal standards. Lipid extracts were aliquoted for quantitation of (dihydro)ceramide/sphingoid bases by LC/MS/MS (3.5 ml) and for normalization of lipids using total phospholipid levels (0.5 ml). The dried lipid aliquots for the analysis of (dihydro)ceramide/sphingoid bases were resuspended in 150 µl of Mobile Phase B composed of MS grade methanol with 0.2% formic acid and 1 mM ammonium formate (pH 5.6) and injected in Vanquish UHPLC Systems (Thermo Fisher Scientific, Waltham, MA, USA) equipped with a Peek Scientific C-8 column (3 µm particle, 4.6 ×150 mm) maintained at 30 °C. HPLC separating conditions are indicated in Supplementary Table 3. Instruments and parameters for mass spectrometry detection are indicated in Supplementary Table 4. Transitions and retention time for sphingolipid species targeted and internal standards used are reported in Supplementary Data 3. Lipids were normalized against nmoles of inorganic phosphate (Pi) derived from total phospholipids in each sample. Dried lipid aliquots set aside from the initial extraction were extracted as described[73] and Pi measurements were performed on the organic lower phase and calculated against a standard curve.

### Discovery lipidomics

2-4 ×$10^6$ cells were harvested, washed with PBS, and frozen as cell pellets. Lipids were extracted and analyzed with LC-MS/MS by National Center for Quantitative Biology of Complex Systems at University of Wisconsin-Madison. For lipid extraction, MeOH (4°C) (225 µL) was added and vortexed (30 s). Methyl tert-butyl ether (750 µL, 4°C) was added and vortexed (30 s). The samples were mixed using an orbital shaker (6 mins) to extract lipids. $H_2O$ (225 µL, 4°C) was added and vortexed (30 s) to induce phase separation. Samples were centrifuged (14,000 g, 2 min, 4°C) to complete phase separation. 200 µL of the organic phase was transferred to a clean vial and dried in a vacuum centrifuge (45 min). The organic residue was reconstituted in MeOH/Toluene (9:1, v/v) (100 µL) for LC-MS/MS analysis. Sample analysis was performed on an Acquity CSH C18 column held at 50 °C (2.1 ×100 mm x 1.7 mm particle size; Waters Corporation) using a Vanquish Binary Pump (400 µL/min; Thermo Scientific). Mobile phase A consisted

of 10 mM ammonium acetate in ACN/$H_2O$ (70:30, v/v) containing 250 µL/L acetic acid. Mobile phase B consisted of 10 mM ammonium acetate in IPA/ACN (90:10, v/v) with the same additives. Initially, mobile phase B was held at 2% for 2 min and then increased to 30% over 3 min. Mobile phase B was then further increased to 85% over 14 min and then raised to 99% over 1 min and held for 7 min. The column was then re-equilibrated for 5 min before the next injection. 10 µL of lipid extract were injected by a Vanquish autosampler (Thermo Scientific). The LC system was coupled to a Q Exactive HF mass spectrometer (Thermo Scientific) by a HESI II heated ESI source (Thermo Scientific). The MS was operated in positive and negative mode (|3500 V|) during sequential injections collecting both $MS^1$ and $MS^2$ spectra. The MS data were processed using Compound Discoverer 2.0 (Thermo Scientific) and LipiDex[74]. The data have been deposited to MassIVE: MSV000092894.

### Statistical analysis

The exact numbers of biological replicates conducted and statistical analysis used for each experiment are stated in the figure legends. No data were excluded from the analysis. Statistical analyses were performed with GraphPad Prism 7. To compare the difference between two groups, unpaired or paired two-tailed Student's t-tests were conducted. For multiple comparisons, one-way ANOVA or two-way ANOVA was used, followed by Tukey's, Dunnett's or Sidak's multiple comparisons test.

### Reporting summary

Further information on research design is available in the Nature Portfolio Reporting Summary linked to this article.

## Data availability

The discovery lipidomics data generated in this study have been deposited to MassIVE database under accession code MSV000092894 [https://massive.ucsd.edu/ProteoSAFe/dataset.jsp?task=e77cdb5b6322473ba9b27b0df992c75d]. All other data generated in this study are provided in the paper or in the Supplementary Information and Source Data files. Source data are provided with this paper.

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

## Acknowledgements

This work was funded by National Institutes of Health Grant DK50107 to E.H.B., the National Center for Quantitative Biology of Complex Systems P41 GM108538 to J.J.C., National Institutes of Health, National Cancer Institute Grant P01 CA097132 to C.L., and Carbone Cancer Center P30 CA014520. We thank Dr. Don Wojchowski (University of New Hampshire) for generously providing anti-EPOR antibody.

## Author contributions

R.L. and E.H.B conceived the study. R.L. performed and analyzed most of the experiments. A.B. performed ceramide sensitivity analysis for erythroid vs. myeloid cells. S.Q. performed RT-qPCR experiment to measure GATA1-dependent induction of *Degs1*. H.H. performed RT-qPCR analysis to measure Cers gene expression. J.J.C and K.A.O performed and analyzed discovery lipidomics experiments. C.L. and Y.A.H performed and analyzed quantitative lipidomics for sphingolipids. R.L. and E.H.B wrote the manuscript.

## Competing interests

J.J.C is a consultant for Thermo Fisher Scientific, 908 Devices, and Seer. The remaining authors declare no competing interests.
