## [Peer Review File · Nature Communications]

A Transcriptional Network Governing Ceramide Homeostasis Establishes a Cytokine-Dependent Developmental ProcessREVIEWER COMMENTS

Reviewer #1 (Remarks to the Author):

Transcriptional Network Governing Ceramide Homeostasis Establishes a Cytokine-Dependent Developmental Process.

Corresponding author. E. Bresnick

The manuscript from Liao et al. describes the role of Ceramide homeostasis in regulating erythropoiesis. The authors identify DES, an enzyme that regulates ceramide and sphingosine biosynthesis, as being a Gata1 target. They go on to show that interrupting ceramide biosynthesis or treating cells with excess C6-ceramide decreases erythroid maturation. Analysis of erythroid differentiation in G1E-ER4 cells and primary mouse fetal liver and human peripheral blood erythroid progenitors show that ceramide regulates signal transduction by affecting the activity of PP2a. In addition, they show that erythropoiesis is marked more sensitive to alterations in ceramide metabolism than myeloid differentiation.

Overall, this is a strong manuscript that identifies a new pathway regulated by Gata1 during erythropoiesis. In addition, their data linking ceramide homeostasis to signaling events downstream of the Epo receptor identifies a novel area of research. I have a few questions about the data that need to be addressed.

1. Treatment of cells with 4HPR leads to changes in levels of Cer and dhCer species (Fig 1 G and H). Do the corresponding ceramide synthase genes change expression as well?
2. Treating G1E-ER4 cells with 4HPR blocks the increase in EpoR expression (Figure 2E). Can you rule out that the effects of 4HPR on erythroid differentiation are not due to defects in EpoR expression?
3. In extended figure 2D, it looks like 4HPR blocks the progression from CD71-Ter119- to CD71+Ter119- cells, is that also true if you treat mouse or human primary erythroid progenitors with 4HPR?
4. Does C6-cer or another ceramide rescue G1E-ER4 cells treated with 4HPR?
5. Treating cells with C6-cer also inhibits erythroid development, does this C6-cer get metabolized into downstream products such as Sphingosine? How do you rule out that the effects of C6-cer are not caused in part by downstream metabolites?
6. The signaling experiments are crucial to the conclusions of the paper, but with the exception of Figure 6K, none of the blots show levels of EpoR, p-EpoR, Kit and p-Kit. Given that 4HPR decrease EpoR expression, these controls should be included.
7. Do you get a shift in the dose response to Epo in colony formation when cultures are treated with C6-Cer?
8. An explanation of how the bacterial sphingomyelinase works would be helpful. There is no description of this experiment in the methods.

Reviewer #2 (Remarks to the Author):

The ceramide pathway was previously shown to inhibit erythropoiesis in various models, including human hematopoietic stem cells, whereas sphingosine-1 phosphate restored erythroid differentiation. Ceramides (Cer) were shown to trigger granulomonocytic differentiation. These events were already correlated to the modulation of GATA-1/2 transcription factors. Moreover, ceramides were shown to modulate the mTOR/AKT pathway leading to inhibition of autophagy required for erythroid differentiation.

Here Liao et al. further extend this knowledge by establishing a link between GATA-1 and the gene battery required for Cer metabolism by using a transcriptome/proteome approach. They conclude that GATA1 regulation of sphingolipid metabolic genes contributes to physiological ceramide homeostasis, essential for erythroid differentiation. They clearly demonstrated that Cer homeostasis is essential for erythroblast function and differentiation. Subsequently, they disrupted Cer homeostasis to yield excessive dhCer or Cer inhibiting erythroid progenitors in line with previously published data. This disruption modulates cytokine signaling systems essential for erythroid progenitor. In addition, they discovered Ormdl3 an upstream regulator of ceramide synthesis. Eventually, they show that Cer interferes with Epo and SCF receptors.

Whereas the paper is interesting in the field of erythroid differentiation, a number of improvements need to be provided.

1. The authors use a well-known mouse G1E cell model for most of their investigations. The authors are asked to validate key experimental results with erythropoietin-differentiated human hematopoietic stem cells to confirm the relevance of the mouse results.

2. Investigation of viability in Figure 2C. The authors are asked to indicate viability in % and not as a relative unit compared to control.

Extended data figure 2A shows FACS results of AnnexinV/PI staining. This analysis is likely artefactual as the voltage in FSC versus SSC was probably erroneously set, leading to a distortion of the result. This representation/experiment needs to be considerably improved/redone.

The same figure shows that the cells were largely dying, even without treatment. 64% of viability as a starting point hints at sensitization of the cells. How do the authors explain this low viability and the impact on the outcome of the treatments?

3. In Extended data Figure 5, the authors observe a significantly reduced SCF-induced p-AKT and Epo-induced p-AKT and p-STAT5. As the WB in the manuscript only shows a modest variation, the authors are asked to present the original unaltered triplicates.

4. Minor: Line 285 and Extended data figures explain the acronym PDBE.

Reviewer #3 (Remarks to the Author):

Comments to the manuscript entitle "Transcriptional Network Governing Ceramide Homeostasis Establishes a Cytokine-Dependent Developmental Process".

The lipidomics methodology, targeted quantification of sphingolipids and untargeted lipidomics, applied in this study are well described and suitable to main purpose of the manuscript. However, in order to help readers and future similar experiments I do recommend to:

-add a brief method description containing the most relevant information for sphingolipids quantification. This information should contain a table with sphingolipid species targeted, retention time, transitions monitored and internal standard use for quantification.

-deposit the data obtained from the untargeted lipidomics approach, including MS and MS/MS spectra, in a public data repository platform like Metabolomics Workbench.

Other comments:

In the excel file provided as attached file there is information about sphingolipids quantification. This information has not been formatted and it is reported as exported from the used quantification software. Only if the readers are familiar with this type of data can be understand. So I do recommend to just keep the summarized table containing the concentration levels. On the other hand, in this table there is only one dihydroceramide (DhCer C16:0) but in Figure 1H the heatmap contains other DhCer species. Where these other DhCer targeted and quantified together with other sphingolipids?

Reviewer #1:

1) **Query:** “Treatment of cells with 4HPR leads to changes in levels of Cer and dhCer species (Fig 1 G and H). Do the corresponding ceramide synthase genes change expression as well?”

Response: We thank the reviewer for pointing out the potential changes in ceramide synthase gene expression upon 4HPR treatment which could lead to changes in Cer and dhCer species. To address this question, we conducted RT-qPCR experiments to quantify the RNA levels of *Cers* genes after 4HPR treatment. Out of 6 *Cers* genes, only *Cers2* and *Cers6* are expressed abundantly in our cell system (Supplementary Fig. 2a). A 24 h 4HPR treatment did not alter the RNA expression of *Cers2* or *Cers6* (Supplementary Fig. 2b), providing evidence that 4HPR disrupts the balance between dhCer and Cer without altering the expression of *Cers* genes.

2) **Query:** “Treating G1E-ER4 cells with 4HPR blocks the increase in *EpoR* expression (Figure 2E). Can you rule out that the effects of 4HPR on erythroid differentiation are not due to defects in *EpoR* expression?”

Response: In the original submission, we demonstrated that a 24 h 4HPR treatment reduced *Epor* expression during erythroid differentiation. To address if the defects in *Epor* expression contribute to 4HPR effects on erythroid cell function, we conducted new experimentation to measure *Epor* and *Kit* expression after a 4 h 4HPR treatment, which strongly reduced cytokine signaling. RT-qPCR revealed that neither *Epor* nor *Kit* expression was altered upon 4 h 4HPR treatment (Supplementary Fig. 4c), suggesting that 4HPR inhibits Epo signaling without impacting *Epor* expression. Analogous to how prolonged SCF/c-Kit signaling contributes to the expression of *Kit* (Zhu et al., *Blood*, 2011), the decreased *Epor* expression after 24 h 4HPR treatment is likely secondary to the 4HPR-mediated inhibition of Epo signaling.

3) **Query:** “In extended figure 2D, it looks like 4HPR blocks the progression from CD71-Ter119- to CD71+Ter119- cells, is that also true if you treat mouse of human primary erythroid progenitors with 4HPR?”

Response: We conducted new experimentation in primary mouse fetal liver cells treated with 4HPR and assessed erythroid differentiation using flow cytometry. Like what we observed in G1E-ER-GATA1 cells, 4HPR increased the percentage of CD71-Ter119-cells and decreased the percentage of CD71+Ter119- cells (Supplementary Fig. 2i and j).

4) **Query:** “Does C6-cer or another ceramide rescue G1E-ER4 cells treated with 4HPR?”

Response: We conducted new experimentation in G1E-ER-GATA1 cells to ask if C6-Cer rescues 4HPR-dependent inhibition of cytokine signaling (Supplementary Fig. 4a and b). C6-Cer single

Faculty

Anita Bhattacharyya, PhD
Barak Blum, PhD
Grace Boekhoff-Falk, PhD
Marjorie Brand, PhD
Emery Bresnick, PhD

Ricki Colman, PhD
Jeffrey Dilworth, PhD
Ying Ge, PhD
Anne Griep, PhD
Timothy Kamp, MD, PhD

Junsu Kang, PhD
Pamela Kreeger, PhD
Youngsook Lee, PhD
Valentina Lo Sardo, PhD
Bo Liu, PhD
Ahmed Mahmoud, PhD

Suzanne Ponik, PhD
Igor Slukvin, MD, PhD
Rupa Sridharan, PhD
Owen Tamplin, PhD
Beth Weaver, PhD
Deneen M. Wellik, PhD

Professors Emeriti:

Karen M. Downs, PhD
Daniel Greenspan, PhD
Colin Jefcoate, PhD
Joseph W. Kemnitz, PhD
Richard L. Moss, PhD
James Thomson, VMD, PhD

treatment or C6-Cer and 4HPR double treatment abrogated Epo-dependent phosphorylation of AKT and ERK. C6-Cer and 4HPR double treatment further reduced Epo-induced p-STAT5 in comparison to a single treatment. These results suggest that ceramide does not rescue the impact of 4HPR on cytokine signaling, further strengthening our conclusion that accumulation of dhCer or Cer is detrimental to erythroid cells.

5) **Query:** *“Treating cells with C6-cer also inhibits erythroid development, does this C6-cer get metabolized into downstream products such as Sphingosine? How do you rule out that the effects of C6-cer are not caused in part by downstream metabolites?”*

Response: C6-Cer can be incorporated into sphingolipid metabolic pathways and metabolized into downstream products, like sphingosine, endogenous long-chain ceramides, and sphingomyelin. We demonstrated that inhibition of cytokine signaling by C6-Cer depends on PP2A activity (Fig. 5), and prior biochemical analyses by the Hannun laboratory demonstrated that either short-chain or long-chain ceramide bind PP2A and enhance its activity *in vitro* (Dobrowsky, et al., *J. Biol. Chem.*, 1993; Chalfant et al., *J. Biol. Chem.*, 1999). Since no other sphingolipid species have been reported to enhance PP2A activity *in vitro*, we believe that the PP2A-dependent C6-Cer activity is primarily due to increased intracellular ceramide levels.

6) **Query:** *“The signaling experiments are crucial to the conclusions of the paper, but except for Figure 6K, none of the blots show levels of EpoR, p-EpoR, Kit and p-Kit. Given that 4HPR decrease EpoR expression, these controls should be included.”*

Response: We thank the reviewer for highlighting the need to analyze KIT and EPOR activation in the signaling experiments. We conducted new experimentation to measure KIT and EPOR phosphorylation upon acute (30 min) or prolonged (4 h) disruption of ceramide homeostasis by 4HPR, C6-dhCer, or C6-Cer. These analyses revealed that a 4 h treatment with 4HPR, C6-dhCer, or C6-Cer reduced SCF-dependent KIT phosphorylation and Epo-dependent EPOR phosphorylation (Fig. 4j-m), while acute C6-Cer treatment reduced Epo-dependent EPOR and JAK2 phosphorylation without impacting p-KIT (Fig. 6k-n). These results provide further evidence for our model that a dual-component mechanism mediates Cer/PP2A activity to regulate cytokine signaling (Fig. 6o).

7) **Query:** *“Do you get a shift in the dose response to Epo in colony formation when cultures are treated with C6-Cer?”*

Response: In Fig. 3c, we demonstrated that C6-Cer almost completely abrogated erythroid colonies (CFU-E and BFU-E) in M3434 media that contains maximal level of Epo. Therefore, one would expect to see little to no erythroid colonies for cultures treated with C6-Cer in response to sub-maximal levels of Epo.

8) **Query:** *“An explanation of how the bacterial sphingomyelinase works would be helpful. There is no description of this experiment in the methods.”*

Response: We thank the reviewer for pointing this out. We have incorporated a description of the bacterial sphingomyelinase experiment in the Methods under Cell lines and primary cultures on page 21, line 490-492.

Summary of major changes implemented:

In addition to textual revisions to address the reviewers' comments, we incorporated the following new data, which reflects the advancement of the science and efforts to further increase rigor and novelty.

1. We conducted new experimentation to quantify the expression of ceramide synthase genes upon 4HPR treatment. A 24 h 4HPR treatment did not change *Cers2* and *Cers6* RNA levels (Supplementary Fig. 2a and b), providing evidence that 4HPR disrupts ceramide homeostasis without altering ceramide synthase expression.
2. We conducted new analyses to establish the time course for 4HPR activity in G1E-ER-GATA1 cells, which revealed a time-dependent loss of cell viability upon 4HPR treatment (Fig. 2c and d).
3. We revised the experimental conditions and repeated the flow cytometry-based apoptotic analyses with G1E-ER-GATA1 cells treated with 4HPR. The new data are incorporated into Supplementary Fig. 2c and d.
3. We conducted new experimentation in primary human erythroblasts to analyze the impact of 4HPR on cell proliferation and survival. Like what we observed in G1E-ER-GATA1 cells, 4HPR significantly reduced cell viability 24 h post-treatment (Fig. 2e) and significantly increased the percentage of early and late apoptotic cells (Supplementary Fig. 2e and f).
4. We conducted new experimentation to analyze the impact of 4HPR on erythroid differentiation using primary murine fetal liver cells (Supplementary Fig. 2i and j).
5. We conducted new experimentation to investigate how disrupting ceramide homeostasis impacts cytokine-dependent receptor activation. These data are incorporated into Fig. 4j-m.
6. We conducted new experimentation to ask if ceramide rescues the adverse impact of 4HPR on cytokine signaling. These data are incorporated into Supplementary Fig. 4a and b.
7. We conducted new experimentation to ask if PP2A inhibition rescues 4HPR-dependent inhibition of signaling. These data are incorporated into Fig. 5a and Supplementary Fig. 5a, which further strengthened our conclusion that disrupting ceramide homeostasis inhibits cytokine signaling via a PP2A-dependent mechanism.
8. We conducted new experimentation in primary human erythroblasts to analyze how disrupting ceramide homeostasis acutely impacts cytokine signaling (Fig. 6i and j). The results reinforced our findings in G1E-ER-GATA1 cells.
9. We conducted new experimentation to analyze how disrupting ceramide homeostasis acutely impacts Epo receptor activation (Fig. 6m-n).

We very much thank the reviewer for providing insightful comments.

Reviewer #2:

1) **Query:** *“The authors use a well-known mouse G1E cell model for most of their investigations. The authors are asked to validate key experimental results with erythropoietin-differentiated human hematopoietic stem cells to confirm the relevance of the mouse results.”*

Response: We thank the reviewer for highlighting the importance of using primary human hematopoietic stem cell differentiation data to compare with the mouse results. In addition to the cytokine signaling experiments with 4HPR, C6-dhCer, and C6-Cer treatment of primary human CD34⁺ cell-derived erythroblasts, which as included in the original submission (Fig. 4g-i), we conducted new experimentation with this primary human erythroblast system. 4HPR treatment of primary human erythroblasts during a 96 h time course revealed that 4HPR strongly reduced cell viability 24 h post-treatment (Fig. 2e). We also conducted flow cytometry-based Annexin V analysis with 4HPR-treated primary human erythroblasts, which revealed that 4HPR strongly increased early and late apoptotic populations (Supplementary Fig. 2e and f). To maximize the opportunity to identify primary mechanistic insights, we treated human erythroblasts with C6-dhCer and C6-Cer acutely, which revealed that an acute disruption of ceramide homeostasis attenuated SCF- and Epo-induced p-AKT, without affecting Epo-induced p-STAT5 (Fig. 6i and j). In aggregate, these results extend and confirm our findings with the mouse G1E-ER-GATA1 cell model, and therefore, these mechanisms are conserved between human and mouse.

2) **Query:** *“Investigation of viability in Figure 2C. The authors are asked to indicate viability in % and not as a relative unit compared to control.*

Extended data figure 2A shows Facs results of AnnexinV/PI staining. This analysis is likely artefactual as the voltage in FSC versus SSC was probably erroneously set, leading to a distortion of the result. This representation/experiment needs to be considerably improved/redone.

The same figure shows that the cells were largely dying, even without treatment. 64% of viability as a starting point hints at sensitization of the cells. How do the authors explain this low viability and the impact on the outcome of the treatments?”

Response: We thank the reviewer for pointing out the potential artifact of the flow data. We repeated these experiments with a revised protocol to mitigate false-positive Annexin V staining. The new data are incorporated into Supplementary Fig. 2c and d. We also conducted the same experiment in primary human erythroblasts (Supplementary Fig. 2e and f), which validated our findings in the mouse system.

We re-analyzed the data in Fig. 2c and expressed the data as viability in %. The same analysis was conducted in Fig. 2d and e.

3) **Query:** *In Extended data Figure 5, the authors observe a significantly reduced SCF-induced p-AKT and Epo-induced p-AKT and p-STAT5. As the WB in the manuscript only shows a modest variation, the authors are asked to present the original unaltered triplicates.”*

Response: We thank the reviewer for highlighting the importance of rigor and reproducibility, which is a critical component of our experimental approach. We incorporated the Western blots from all four biological replicates into Supplementary Fig. 5d, which reveals the rigor and reproducibility of the data.

4) **Query:** *“Minor: Line 285 and Extended data figures explain the acronym PBDE.”*

Response: We defined the acronym PBDE (peripheral blood-derived erythroblasts) when it first appears in the text on page 7, line 147. We also added an explanation for PBDE in the figure legends.

Summary of major changes implemented:

In addition to textual revisions to address the reviewers' comments, we incorporated the following new data, which reflects the advancement of the science and efforts to further increase rigor and novelty.

1. We conducted new experimentation to quantify the expression of ceramide synthase genes upon 4HPR treatment. A 24 h 4HPR treatment did not change *Cers2* and *Cers6* RNA levels (Supplementary Fig. 2a and b), providing evidence that 4HPR disrupts ceramide homeostasis without altering ceramide synthase expression.

2. We conducted new analyses to establish the time course for 4HPR activity in G1E-ER-GATA1 cells, which revealed a time-dependent loss of cell viability upon 4HPR treatment (Fig. 2c and d).

3. We revised the experimental conditions and repeated the flow cytometry-based apoptotic analyses with G1E-ER-GATA1 cells treated with 4HPR. The new data are incorporated into Supplementary Fig. 2c and d.

3. We conducted new experimentation in primary human erythroblasts to analyze the impact of 4HPR on cell proliferation and survival. Like what we observed in G1E-ER-GATA1 cells, 4HPR significantly reduced cell viability 24 h post-treatment (Fig. 2e) and significantly increased the percentage of early and late apoptotic cells (Supplementary Fig. 2e and f).

4. We conducted new experimentation to analyze the impact of 4HPR on erythroid differentiation using primary murine fetal liver cells (Supplementary Fig. 2i and j).

5. We conducted new experimentation to investigate how disrupting ceramide homeostasis impacts cytokine-dependent receptor activation. These data are incorporated into Fig. 4j-m.

6. We conducted new experimentation to ask if ceramide rescues the adverse impact of 4HPR on cytokine signaling. These data are incorporated into Supplementary Fig. 4a and b.

7. We conducted new experimentation to ask if PP2A inhibition rescues 4HPR-dependent inhibition of signaling. These data are incorporated into Fig. 5a and Supplementary Fig. 5a, which further strengthened our conclusion that disrupting ceramide homeostasis inhibits cytokine signaling via a PP2A-dependent mechanism.

8. We conducted new experimentation in primary human erythroblasts to analyze how disrupting ceramide homeostasis acutely impacts cytokine signaling (Fig. 6i and j). The results reinforced our findings in G1E-ER-GATA1 cells.

9. We conducted new experimentation to analyze how disrupting ceramide homeostasis acutely impacts Epo receptor activation (Fig. 6m-n).

We very much thank the reviewer for providing insightful comments.

Reviewer #3:

1) **Query:** *“-add a brief method description containing the most relevant information for sphingolipids quantification. This information should contain a table with sphingolipid species targeted, retention time, transitions monitored and internal standard use for quantification.”*

Response: We updated methods relevant to quantitative lipidomics for sphingolipids on pages 23-24, line 551-566, and provided additional information on the HPLC separating conditions, instruments and parameters for mass spectrometry detection, and transitions and retention time for sphingolipid species targeted and internal standards used in Supplementary Table 5-7.

2) **Query:** *“-deposit the data obtained from the untargeted lipidomics approach, including MS and MS/MS spectra, in a public data repository platform like Metabolomics Workbench.”*

Response: We deposited the discovery lipidomics data into MassIVE: MSV000092894. Reviewers can access with the following login information:

user: MSV000092894_reviewer

password: reviewer_pw

3) **Query:** *“In the excel file provided as attached file there is information about sphingolipids quantification. This information has not been formatted and it is reported as exported from the used quantification software. Only if the readers are familiar with this type of data can be understand. So I do recommend to just keep the summarized table containing the concentration levels. On the other hand, in this table there is only one dihydroceramide (DhCer C16:0) but in Figure 1H the heatmap contains other DhCer species. Where these other DhCer targeted and quantified together with other sphingolipids?”*

Response: We thank the reviewer for pointing out the complexity of the Excel file for sphingolipid quantitation. As the reviewer suggested, we updated Supplementary Table 1 to only show the normalized concentration (pmole/nmole Pi) of lipids. In addition, we incorporated the quantitation for all dhCer species in Supplementary Table 1.

Summary of major changes implemented:

In addition to textual revisions to address the reviewers' comments, we incorporated the following new data, which reflects the advancement of the science and efforts to further increase rigor and novelty.

1. We conducted new experimentation to quantify the expression of ceramide synthase genes upon 4HPR treatment. A 24 h 4HPR treatment did not change *Cers2* and *Cers6* RNA levels (Supplementary Fig. 2a and b), providing evidence that 4HPR disrupts ceramide homeostasis without altering ceramide synthase expression.

2. We conducted new analyses to establish the time course for 4HPR activity in G1E-ER-GATA1 cells, which revealed a time-dependent loss of cell viability upon 4HPR treatment (Fig. 2c and d).

3. We revised the experimental conditions and repeated the flow cytometry-based apoptotic analyses with G1E-ER-GATA1 cells treated with 4HPR. The new data are incorporated into Supplementary Fig. 2c and d.

3. We conducted new experimentation in primary human erythroblasts to analyze the impact of 4HPR on cell proliferation and survival. Like what we observed in G1E-ER-GATA1 cells, 4HPR significantly

reduced cell viability 24 h post-treatment (Fig. 2e) and significantly increased the percentage of early and late apoptotic cells (Supplementary Fig. 2e and f).

4. We conducted new experimentation to analyze the impact of 4HPR on erythroid differentiation using primary murine fetal liver cells (Supplementary Fig. 2i and j).

5. We conducted new experimentation to investigate how disrupting ceramide homeostasis impacts cytokine-dependent receptor activation. These data are incorporated into Fig. 4j-m.

6. We conducted new experimentation to ask if ceramide rescues the adverse impact of 4HPR on cytokine signaling. These data are incorporated into Supplementary Fig. 4a and b.

7. We conducted new experimentation to ask if PP2A inhibition rescues 4HPR-dependent inhibition of signaling. These data are incorporated into Fig. 5a and Supplementary Fig. 5a, which further strengthened our conclusion that disrupting ceramide homeostasis inhibits cytokine signaling via a PP2A-dependent mechanism.

8. We conducted new experimentation in primary human erythroblasts to analyze how disrupting ceramide homeostasis acutely impacts cytokine signaling (Fig. 6i and j). The results reinforced our findings in G1E-ER-GATA1 cells.

9. We conducted new experimentation to analyze how disrupting ceramide homeostasis acutely impacts Epo receptor activation (Fig. 6m-n).

We very much thank the reviewer for providing insightful comments.

REVIEWERS' COMMENTS

Reviewer #1 (Remarks to the Author):

The authors have addressed my concerns.

Reviewer #2 (Remarks to the Author):

Based on the additions and improvements the authors provided, I agree with the revised manuscript as presented.

Reviewer #3 (Remarks to the Author):

Authors have addressed all the comments and modify the manuscript accordingly.